# HypoRiPPAtlas as an Atlas of hypothetical natural products for mass spectrometry database search

Yi-Yuan Lee [1,2,7], Mustafa Guler [1,7], Desnor N. Chigumba[3], Shen Wang[1], Neel Mittal[1], Cameron Miller[1], Benjamin Krummenacher[1], Haodong Liu[1], Liu Cao [1], Aditya Kannan [1], Keshav Narayan[1], Samuel T. Slocum[4], Bryan L. Roth [4], Alexey Gurevich [5,6], Bahar Behsaz[1], Roland D. Kersten[3] & Hosein Mohimani [1] ✉

Recent analyses of public microbial genomes have found over a million biosynthetic gene clusters, the natural products of the majority of which remain unknown. Additionally, GNPS harbors billions of mass spectra of natural products without known structures and biosynthetic genes. We bridge the gap between large-scale genome mining and mass spectral datasets for natural product discovery by developing HypoRiPPAtlas, an Atlas of hypothetical natural product structures, which is ready-to-use for in silico database search of tandem mass spectra. HypoRiPPAtlas is constructed by mining genomes using seq2ripp, a machine-learning tool for the prediction of ribosomally synthesized and post-translationally modified peptides (RiPPs). In HypoRiPPAtlas, we identify RiPPs in microbes and plants. HypoRiPPAtlas could be extended to other natural product classes in the future by implementing corresponding biosynthetic logic. This study paves the way for large-scale explorations of biosynthetic pathways and chemical structures of microbial and plant RiPP classes.

The natural products of cultured microbes have served as a major source of lead compounds for antibiotics[1], drug[2], food preservative[3], and analgesic agent[4,5] discoveries. However, antibiotics with diverse modes of actions are needed to combat antibiotics resistance, and a continued focus on the abundant molecules from cultured microbes is ineffective due to high rates of rediscovery. Traditional approaches rely on repeated fractionation and bioactivity testing, followed by isolation and structure elucidation of the molecules of interest, which is a time-consuming and expensive process. The Synthetic-Bioinformatic Natural Products (syn-BNPs)[6], proposed as an alternative strategy, relies on predicting chemical structures with existing bioinformatic tools, and thus, its effectiveness is constrained by the limitations of these tools.

During the past decade, two distinct breakthroughs have revolutionized the field of natural product discovery. First, in the 2000's genome mining approaches made it possible to predict the biosynthetic gene clusters (BGCs) of natural products from microbial DNA sequences[7]. Later, machine learning approaches enabled partial prediction of the building blocks of natural products from their BGCs[8–10]. Recently, several repositories have been developed containing millions of natural product BGCs extracted from hundreds of thousands of microbial genomes[11–13]. However, connecting these BGCs to their molecular products has not kept pace with the speed of microbial genome sequencing. For example, IMG-ABC reports that out of 411,007 BGCs discovered from microbial genomes in public

[1]Carnegie Mellon University, Pittsburgh, PA 15213, USA. [2]Cornell University, Ithaca, NY 14850, USA. [3]Department of Medicinal Chemistry, University of Michigan, Ann Arbor, MI, USA. [4]Department of Pharmacology, University of North Carolina, Chapel Hill, NC, USA. [5]Helmholtz Institute for Pharmaceutical Research Saarland (HIPS), Helmholtz Centre for Infection Research, Saarbrücken, Germany. [6]Department of Computer Science, Saarland University, Saarbrücken, Germany. [7]These authors contributed equally: Yi-Yuan Lee, Mustafa Guler. ✉e-mail: hoseinm@andrew.cmu.edu

repositories, less than 0.3% (1285) are connected to their molecular products[11]. Existing genome mining approaches usually only predict partial structures of natural products and their monomers[10,14], accurate methods for the prediction of complete structures of natural products are not available.

Since 2015, global natural product social (GNPS) molecular networking infrastructure has brought together over two thousand mass spectral datasets from over five hundred principal investigators containing over seven hundred thousand samples obtained from microbial isolates, host-oriented and environmental communities[15]. Accompanied with molecular networking[16] (a network of mass spectra, where similar spectra are connected with an edge), GNPS is a valuable resource for future natural product discovery. However, over 98% of the billion mass spectra currently stored at GNPS represent the 'dark matter of metabolomics'[17] since all attempts to interpret them have been failed. This 'dark matter' likely consists of spectra of unknown molecules produced by BGCs encoded in existing genomic repositories.

It is challenging to directly link spectra from GNPS to BGCs from IMG-ABC[11], antiSMASH-db[12] and BiG-SLiCE[13], as predictions of natural

product (NP) structures from BGCs remain difficult due to knowledge gaps in NP biosynthesis. To bridge this gap, we present HypoRiPPAtlas, a repository of hypothetical ribosomally synthesized and post-translationally modified peptides (RiPPs) predicted from microbial BGCs (Fig. 1). RiPPs are a group of peptidic natural products with highly diverse structures, functionalities, and bioactivities[4]. RiPPs are usually synthesized as short precursor peptides (below 200 amino acids) consisting of a leader peptide, a core peptide, and a follower peptide. Then, the core peptide is post-translationally modified by tailoring enzymes usually present in a cluster around the precursor peptide.

To populate the HypoRiPPAtlas, we developed seq2ripp, a machine-learning tool for predicting the complete chemical structure of mature RiPPs from genomic data. Seq2ripp contains four modules for doing this. Starting from a microbial genome, genome2bgc, bgc2orf and orf2core modules predict RiPP BGCs, precursor peptides and core peptides, respectively. Then, based on the tailoring enzymes present in the BGC, the core2ripp module predicts a combinatorial list of feasible mature RiPP structures for each core peptide (Fig. 2). Machine learning methods, including profile hidden Markov models[18]

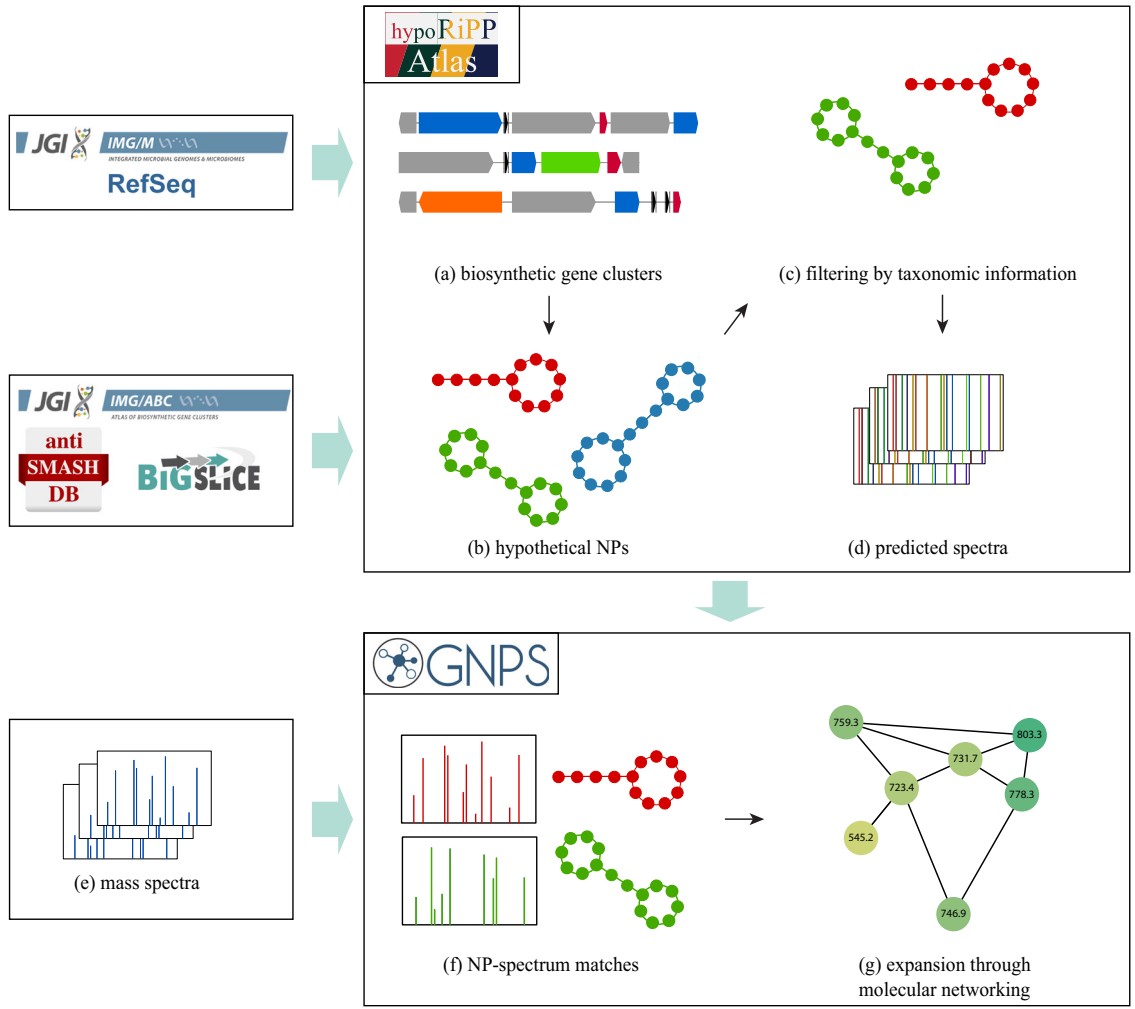

**Fig. 1 | The pipeline for RiPP discovery by HypoRiPPAtlas. a** The pipeline extracts BGCs from microbial genomes available at RefSeq and IMG/M[11], or uses the readily available BGCs from IMG-ABC[11], antiSMASH database[12], and BiG-SLiCE[13]. Colored boxes represent annotated domains. **b** HypoRiPPAtlas is constructed by predicting the hypothetical molecule structures from the BGCs. Three hypothetical structures are distinguished and labeled using different colors. **c** The Atlas is further filtered down to specific taxonomies/gene clusters based on the taxa/metagenomic information available from the samples of interest. **d** Mass spectrometry fragmentation of the hypothetical molecules in the Atlas are predicted, along with known RiPPs from PubChem[74] and NP-atlas[75]. **e** Mass spectra are collected on the environmental samples/microbial isolates, e.g. from a GNPS dataset[15]. **f** Mass spectra are searched against the predicted spectra of hypothetical molecules, and high-scoring RiPP-spectrum matches are reported. **g** The identifications are further expanded through propagation in the molecular network[15]. Steps (**a–d**) are done only once and stored in a repository, while steps (**e–g**) are repeated for every new mass spectral dataset.

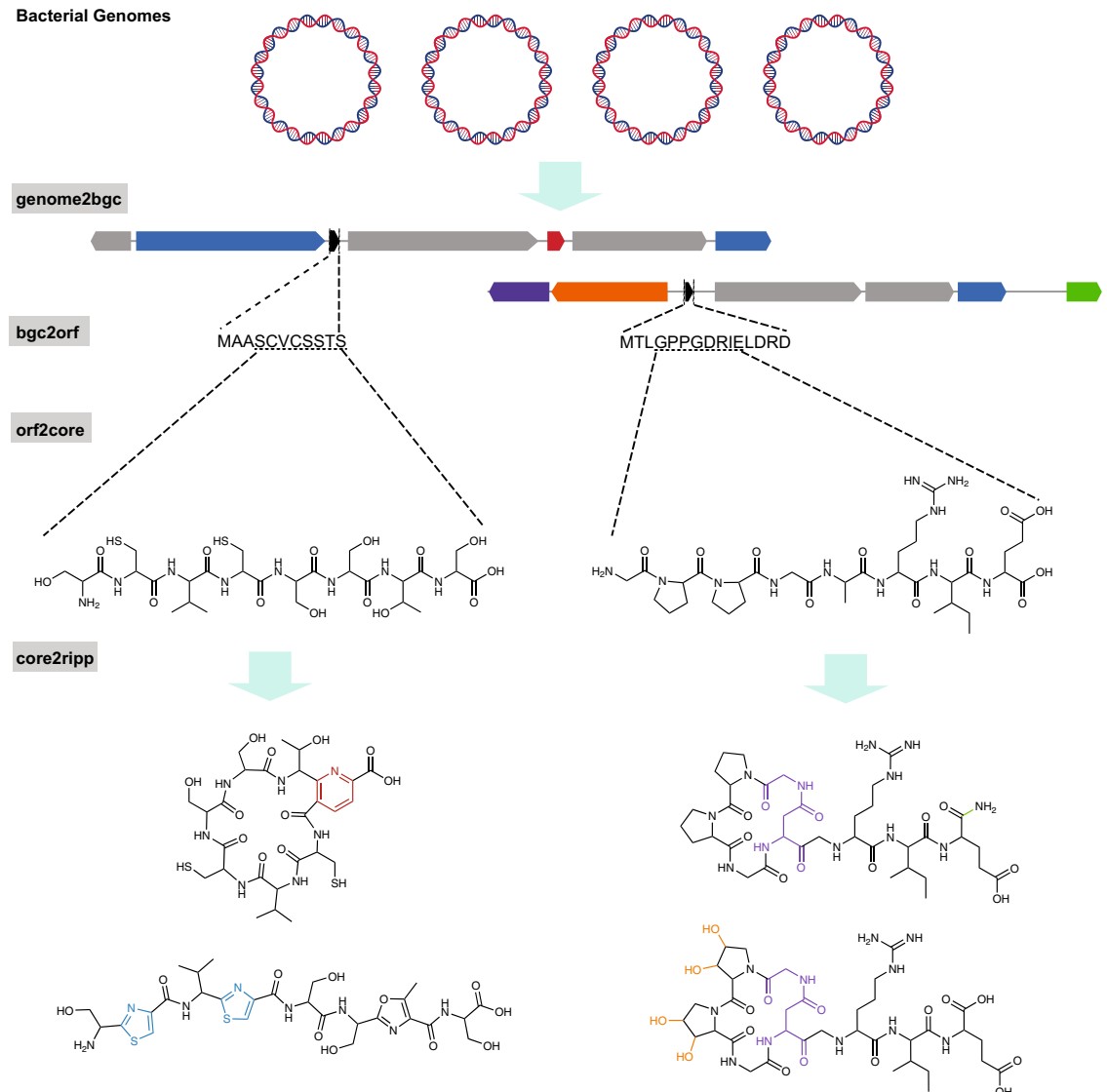

**Fig. 2 | Seq2ripp pipeline.** The process of generating hypothetical RiPPs from genomes in seq2ripp involves four steps: genome2bgc, bgc2orf, orf2core and core2ripp. Genome2bgc extracts BGCs from the input genome by searching RiPP-related genes. Bgc2orf extracts all ORFs from each BGC and identifies the RiPP precursor ORFs. Orf2core identifies the potential cleavage site within the ORFs and produces cores. Core2ripp generates hypothetical chemical structures of the RiPP from the core and the enzymes in the BGC. For example, the colored region in each compound indicates the modification introduced by the enzyme with the corresponding color in the BGC.

and deep neural networks (DNNs), are used in the development of genome2bgc, bgc2orf and orf2core. An efficient subgraph isomorphism algorithm has been developed for predicting the modification sites in core2ripp allowing for applying post-translational modifications. By predicting multiple hypothetical structures, seq2ripp increases the chance of capturing correct mature RiPPs. HypoRiPPAtlas relies on seq2ripp for prediction of the small molecule structure of RiPPs for subsequent generation of their hypothetical mass spectra.

Existing peptidogenomics strategies and tools support a subset of features provided by the seq2ripp pipeline. MetaMiner[19] is limited to modeling RiPPs as strings of amino acids, which is considerably less sensitive than a graph-based representation. RODEO[20] and its updated version, RODEO2[21], predict precursor and core peptides using motif search and machine learning for lassopeptides, class I-IV lanthipeptides, sactipeptides/ranthipeptide, graspetide, linaridin, pyritide, and thiopeptides. RiPPER[22] builds upon RODEO outputs to predict class-independent RiPP precursors by adding ORF prediction via a custom build of gene prediction software Prodigal[23]. PRISM 4[24],

like seq2ripp, is a full pipeline for prediction of RiPP structures from genomic information, but does not include mass spectral analysis for validation of predictions. DeepRiPP[25] predicts RiPP precursors and core peptides with generative models and recurrent neural networks, respectively. NeuRiPP[26] predicts RiPP precursors with convolutional and recurrent neural networks. RiPPminer[27] predicts RiPP precursors, RiPP classes, cleavage sites, and cross-links with support-vector machines and random forest classifiers. The Hypothetical Structure Enumeration and Evaluation (HSEE)[28] method similarly operates on molecular structures and utilizes mass spectrometry for locating post-translational modifications. However, this method does not automatically extract post-translational modifications and requires external tools for generating the theoretical spectra. None of these methods provide a comprehensive pipeline to extract hypothetical molecular structures that aren't tied to a particular subclass of RiPPs.

HypoRiPPAtlas includes hypothetical RiPPs predicted from 22,671 complete microbial genomes. HypoRiPPAtlas reports on hypothetical

BGCs, ORFs, and core sequences extracted from a genomic input. In addition to these sequences, it predicts mature RiPP structures for each identified RiPP BGC. Searching 46 mass spectral datasets from GNPS against HypoRiPPAtlas resulted in the discovery of numerous known and novel RiPPs, including two lassopeptides and one lanthipeptide from *Streptomyces* sp. NRRL B-2660, WC-3904 and WC-3560. Moreover, ribosomal peptides discovered from the human microbiota showed high affinity against human G-protein-coupled receptors (GPCRs) showing the potential of HypoRiPPAtlas in bioactive RiPP identification. Finally, a RiPP class with a newly confirmed post-translational modification (PTM) was characterized by HypoRiPPAtlas from the silverberry plant showcasing that unrevealed RiPP classes can be discovered by our platform.

## Results

### Outline of natural product discovery by HypoRiPPAtlas

Figure 1 illustrates the pipeline for natural product discovery using HypoRiPPAtlas that includes the following steps detailed in the Methods section: (a) BGCs are identified from IMG-ABC/antiSMASH-db/BiG-SLiCE, (b) hypothetical natural products are pre-calculated by seq2ripp in the case of RiPP BGCs. (c) the atlas can be filtered using specific taxonomic information, and (d) spectra for these molecules can be predicted using Dereplicator+[29]. (e) Predicted spectra can be searched against mass spectral datasets using Dereplicator+, yielding (f) molecule-spectrum matches. (g) These matches can be expanded using molecular networking[15]. (h) Links are automatically added between GNPS, HypoRiPPAtlas, and IMG-ABC/BiG-SLiCE.

### Outline of the seq2ripp algorithm

Figure 2 illustrates the seq2ripp pipeline that includes the following steps described in the Methods section: (i) genome2bgc identifies RiPP BGCs based on the biosynthetic enzymes from a microbial genome sequence, (ii) bgc2orf identifies RiPP precursor ORFs from a BGC, (iii) orf2core identifies RiPP core peptides from an ORF, and (iv) core2ripp generates a combinatorial list of feasible mature RiPPs for each core peptide based on the tailoring enzymes present in the BGC.

Genome2bgc extracts RiPP BGCs in the following steps: (i) the genome sequence is translated in six frames, (ii) RiPP-related proteins are identified using hmmsearch[18], (iii) contigs are defined by extracting genome sequence from the middle of the protein to 10,000 bp upstream and downstream, (iv) BGCs are identified after merging overlapping contigs.

Bgc2orf identifies RiPP precursor ORFs in the following steps: (i) the DNA sequence of the BGC is translated in six frames, (ii) RiPP biosynthetic enzymes are identified using hmmsearch[18], (iii) ORFs in the vicinity of the biosynthetic enzymes are extracted (default 10,000 bp), (iv) candidate RiPP precursors are identified using ORF prediction tools[30], and (v) bgc2orf filters this list of candidate ORFs using a deep neural network (see Fig. 3).

Orf2core predicts RiPP core peptides from their ORFs in the following steps: (i) top $k$ N-terminal and C-terminal cleavage sites from each ORF are identified using a deep neural network ($k$ is a user-defined threshold), (ii) a combinatorial list of putative core sequences (up to $k^2$ cases) is generated, and (iii) when the precursor contains repetitive patterns (e.g. cyanobactins[31,32] and plant RiPPs[33-36]), a repeat-specific core finding strategy is used to identify core sequences from repeated leader and follower patterns.

### Outline of the Dereplicator+ algorithm

The Dereplicator+[29] model constructs a theoretical fragmentation of a molecule and scores the predicted theoretical spectrum against a given experimental spectrum. Starting at the original molecule, Dereplicator+ first disconnects all C-C, O-C, and N-C bonds. The resulting connected components are condensed into nodes and re-connected with the previously broken bonds, resulting in the intermediate graph,

referred to as the metabolite graph. Theoretical fragments are defined as the connected components resulting from removing a bridge or a 2-cut in the metabolite graph. Fragments are computed recursively starting at the original metabolite graph and repeating for any child fragments until a user-specified max fragment depth. A theoretical fragment is considered annotated if its corresponding mass can be explained by an experimental peak. A path in the fragmentation graph is considered a fully annotated path if it begins at the root fragment and consists of only annotated fragments. The final score is the number of peaks in the experimental spectrum that belong to fully annotated paths in the theoretical spectrum.

### Datasets

22,671 complete microbial genomes from RefSeq and 2002 draft *Streptomyces* microbial genomes were used for constructing the Atlas. We further analyzed the 46 paired datasets of spectra and genomic data from the Paired Omics Data Platform (PoDP) (917 strains, 7,604,198 MS/MS scans)[37], and a paired dataset of *Actinomyces* (119 strains, 409,245 MS/MS scans, MSV000083738.)

### Identification of radamycin, grisemycin and lacticin 481

We first illustrate the performance of seq2ripp pipeline on radamycin, grisemycin, and lacticin 481. Radamycin is a thiopeptide from *Streptomyces globisporus* isolated from tomato flowers[38,39]. Grisemycin is a linaridin from *Streptomyces griseus* IFO 13350[40]. Lacticin 481 is a lanthipeptide from *Lactococcus lactis* subsp. *lactis*[41]. Radamycin acts as a signaling peptide to regulate the gene expression in *Streptomyces lividans*[39,42]. There is no antimicrobial activity detected in grisemycin[40], whereas lacticin 481 shows antibacterial activity[43].

Genome2bgc identifies 20 hypothetical RiPP BGCs in *Streptomyces globisporus* NRRL B-2709, including the radamycin BGC[44] (Supplementary Data). Bgc2orf further finds 48 ORFs in these BGCs, and orf2core identifies 273 cores in these ORFs. Finally, core2ripp discovers 120,701 hypothetical mature RiPPs in this strain, including the correct radamycin RiPP. Searching mass spectra collected on the extracts of *Streptomyces globisporus* against these 120,701 RiPPs using Dereplicator+ results in a top scoring match with score 25 and p-value $3.0 \times 10^{-46}$ that corresponds to the correct radamycin structure (Fig. 4 and Fig. 5). MetaMiner, which ignores fragmentations between O-C and C-C bonds and higher fragmentation depths, assigns a score of 9 and a p-value of $3.0 \times 10^{-17}$ to the correct radamycin. NeuRiPP[26] and DeepRiPP[25] are both able to identify the correct radamycin ORF. However, DeepRiPP can not identify the correct core sequence (NeuRiPP currently does not contain a core-finding module). Similarly, for grisemycin and lacticin 481, seq2ripp identifies the correct structure among top predictions, and Dereplicator+ correctly picks the correct structure as the most significant match to their mass spectra (Supplementary Figures 1–4).

### Mining microbial genomes

Genome2bgc found 328,676 hypothetical RiPP BGCs in 22,671 microbial genomes. Bgc2orf found hypothetical 55,100 ORFs in these BGCs. Orf2core found hypothetical 1,207,991 cores in these ORFs. Figure 6 summarizes the number of hypothetical BGCs, ORFs and cores retrieved by different seq2ripp modules. DeepRiPP extracts more ORFs than bgc2orf, while NeuRiPP extracts fewer ORFs. DeepRiPP extracts fewer cores than orf2core leading to a larger number of predicted core peptides by seq2ripp modules compared to DeepRiPP across the sampled genomes. The BLAST method, which finds regions of local similarity between potential and known ORFs, extracts the fewest number of ORFs and cores, while the exhaustive method, which extracts all the substrings of the ORFs with lengths ranging from 3 to 30 amino acids, results in the largest amount of ORFs and cores[45]. Unfortunately, it is not possible to compare the sensitivity and specificity of different methods on a large scale.

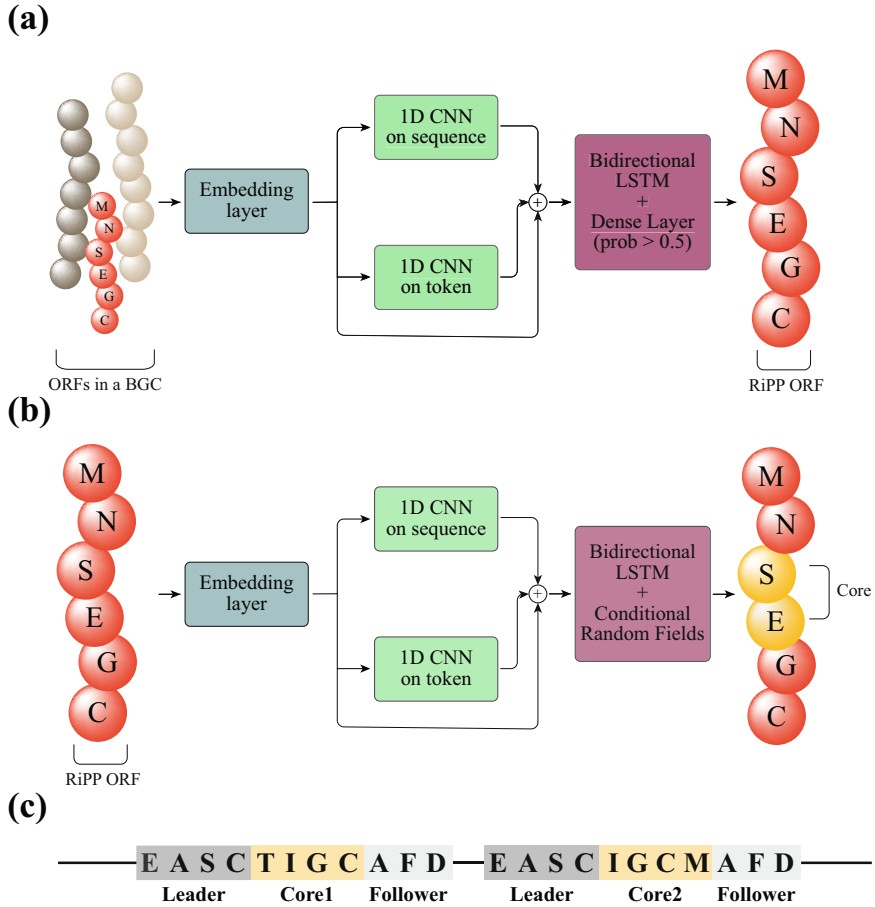

**Fig. 3 | Bgc2orf and orf2core models.** As illustrated in (**a**) and (**b**), from left to right, the red peptide is a RiPP ORF, and the yellow section is the RiPP core. The green blocks are two 1D CNNs, and the purple blocks are bidirectional LSTM with a dense layer and a CRF layer in bgc2orf and orf2core, respectively. **a** Bgc2orf model is a binary classifier that computes the probability of a given ORF peptide sequence being a RiPP ORF. Bgc2orf model consists of (1) a padding process and an embedding layer (shown in blue), (2) two 1D CNNs (shown in green), and (3) a single layer bidirectional LSTM, a flattening layer, and a dense layer (shown in purple). The output is a probability and the default cutoff is 0.5. **b** The orf2core model shares a similar architecture with bgc2orf. However, the flattening and dense layers are replaced with a conditional random fields layer (shown in purple), which predicts the probability of each amino acid is one of the $<start>$, $<before>$, $<core>$, $<after>$, $<end>$ tokens. The orf2core model takes a RiPP ORF as input and identifies $k$ N-terminal and $k$ C-terminal cleavage sites given the predicted tokens, where $k$ is a user-defined hyperparameter. N- and C-terminal cleavage sites are defined as the transition from $<before>$ to $<core>$ and from $<core>$ to $<after>$, respectively. Then, cores are predicted based on the combination of N- and C-terminal cleavage sites. **c** An alternative core finder is used to search the repeated leader-follower patterns, which are highlighted in gray, and to identify the core sequence in the patterns, highlighted in yellow. The alternative core finding is enabled for cyano-bacteria BGCs (which contain the YcaO gene motif) and plants.

## Identifying RiPPs from the PoDP datasets

Analyzing 46 PoDP datasets with 1036 genomes, seq2ripp predicts 17,505 hypothetical BGCs, 54,605 hypothetical ORFs, 118,052 hypothetical cores and 30,687,610 unique hypothetical RiPPs (Fig. 6). After searching these RiPPs against corresponding spectra with Dereplicator+, three RiPPs are identified (Fig. 7).

**Lasso-1648** is identified from *Streptomyces* NRRL B-2660, containing a N-terminal macrolactam ring between $N_1$ and $D_8$ (Fig. 7a and Supplementary Figure 5). Based on Seq2ripp predictions, the PTM is applied by Asn-synthetase (PF00733.18). **Lasso-1795** is identified from *Streptomyces* NRRL B-2660 and WC-3560, containing a N-terminal macrolactam ring between $Q_1$ and $D_8$ (Fig. 7b and Supplementary Figure 6). The PTM is applied by Asn-synthetase (PF00733.18). **Lanthi-1794** is identified from *Streptomyces* WC-3904. A dehydroalanine (Dha) located at $S_6$, and three dehydrobutyrines (Dhb) located in $T_2$, $T_{10}$, $T_{15}$ are potentially connected to one of the cysteines in the core peptide, forming lanthionine (Lan) or methyl-lanthionine (MeLan) rings (Fig. 7c and Supplementary Figure 7). The PTM is applied by Lantibiotic dehydratase (PF05147.6

and PF04738.6). The DTGHCSGVCTVLVCTVAVC core identified by seq2ripp for **lanthi-1794** does not appear in the survey conducted by Walker et al.[46]. However, the precursors and cores for **lasso-1648** and **lasso-1795** were previously reported by RODEO[20]. Seq2ripp validated that these lassopeptides are expressed naturally by the producing microorganisms through mass spectral search.

We searched mass spectral datasets from the PoDP database against the corresponding RiPP molecules from HypoRiPPAtlas using Dereplicator+[29]. At a false discovery rate (FDR) of 1% (score threshold of 15), 64 unique RiPPs (131 molecule-spectrum matches) were discovered. Supplementary Figure 8 shows the number of peptide-spectrum matches (PSMs) and unique peptides identified at different score thresholds in target and decoy.

## Identifying RiPPs from 2002 draft *Streptomyces* microbial genomes

Seq2ripp found 48,542 hypothetical RiPP BGCs, 86,562 hypothetical ORFs, and 2,159,946 hypothetical cores in 2,002 *Streptomyces* draft microbial genomes (Fig. 6).

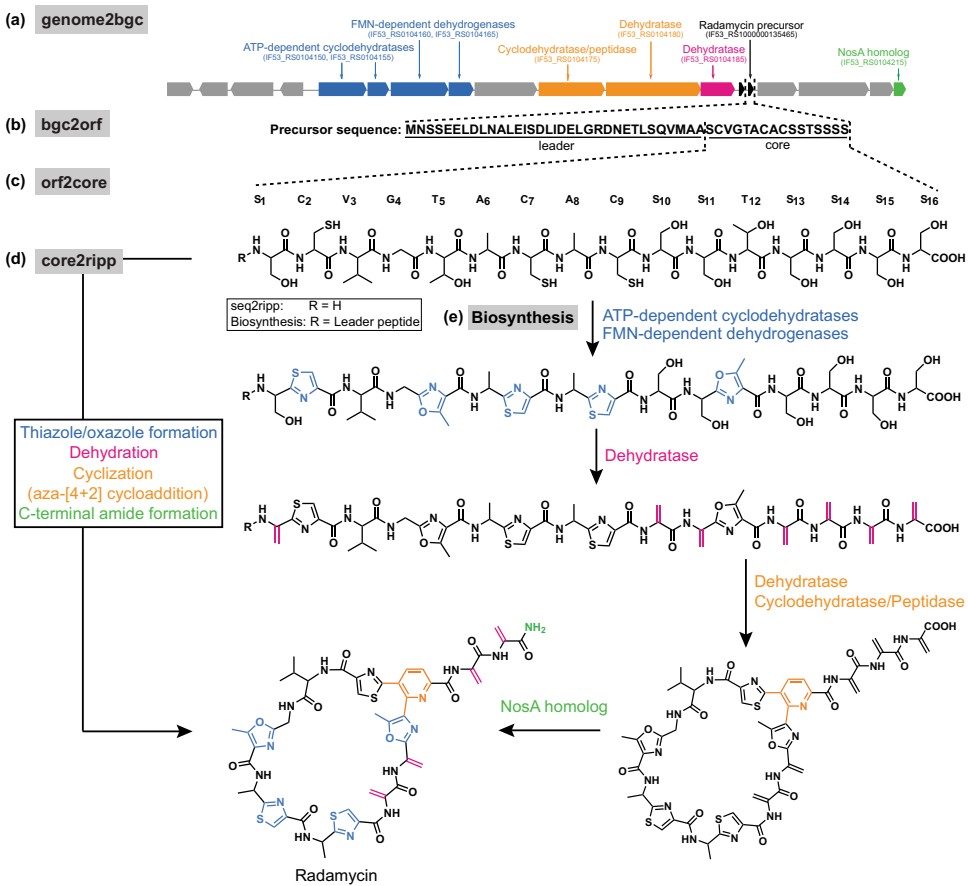

**Fig. 4 | Rediscovery of radamycin by seq2ripp. a** BGCs are identified through their modification enzymes. For example, dehydration enzymes are frequently found in some classes of RiPPs. **b** Short ORFs within 10,000 bp of these enzymes are detected as candidate structural ORFs. **c** Fragments of the structural ORFs with lengths between 5 to 30 amino acids are extracted as candidate precursor peptides. **d** Depending on the tailoring enzymes in the BGC, corresponding modifications are (optionally) applied to the precursor peptides to form hypothetical molecules. **e** seq2ripp prediction is informed by biosynthetic knowledge of thiopeptide.

## Benchmarking bgc2orf

We compared bgc2orf with DeepRiPP and NeuRiPP, state-of-the-art tools for RiPP ORF identification. Since DeepRiPP and NeuRiPP are pre-trained with different datasets, two separate experiments were conducted to compare the prediction accuracy between bgc2orf and DeepRiPP/NeuRiPP on the MIBiG[47] dataset. In the first experiment, we discarded MIBiG RiPPs that were used in the DeepRiPP training data, and in the second experiment, we discarded MIBiG RiPPs that were present in the NeuRiPP training data. In the first experiment, bgc2orf achieved 75.0% accuracy in comparison to 70.0% for DeepRiPP, while in the second experiment, bgc2orf achieved 83.3% accuracy, in comparison to 68.4% for NeuRiPP. Supplementary Figure 9 shows the loss function of bgc2orf network based on the number of epochs. Supplementary Figure 10 shows the re-sampling strategy to avoid data imbalance.

## Benchmarking orf2core

Orf2core generates top $k$ cleavages for each ORF, where $k$ is a user-adjusted parameter. Supplementary Figure 11 shows the tradeoff between accuracy and the number of predicted core peptides for selecting $k$. Moreover, orf2core includes a repeat-finder module for the identification of cores with repetitive patterns (e.g. cyanobactins and plant RiPPs). On a test dataset of 165 cores from MIBiG database (excluding training data of DeepRiPP and orf2core), orf2core correctly identified 48.48% of cores, in comparison to 35.00% for DeepRiPP. When the top 5 pairs of cleavage sites were allowed, orf2core correctly identified 63.03% of cores. Supplementary

Figure 12 shows the loss of orf2core network based on the number of epochs.

## Cross-validation (CV) of bgc2orf and orf2core

10-fold CV was conducted to estimate the performance of bgc2orf and orf2core. Datasets for each model were split into ten subsets. In each fold, one subset was used for the test and the other nine subsets were merged for training and validation. Each subset was only used for the test once in the 10-fold CV. 10-fold CV provides a better estimation of the model performance, especially when training data is limited. The average test accuracies for bgc2orf and orf2core are 94.20% and 73.31%, respectively. The test accuracy among 10-fold CV in each model is consistent, indicating that each model is generalized to all available data we can collect at the time of the experiment (Supplementary Figure 13).

## A comparison of RiPP discovery methods

We compared seq2ripp's modules against the following computational tools identified by Russel and Truman, 2020[48] and described in Fig. 8a.

BAGEL4[49] uses manually-curated HMMs to detect and annotate RiPP BGCs and ORFs. AntiSMASH6[50], though not specific to RiPP biosynthesis, predicts and characterizes RiPP BGCs, including the identification of putative precursor and core peptides for some RiPP classes using manually-curated HMMs in conjunction with tools such as KnownClusterBlast[51] and RODEO[20]. RiPPMiner[27] predicts chemical structures from precursor peptides for select classes of RiPPs using Support Vector Machines. NeuRiPP[26] predicts genuine precursor

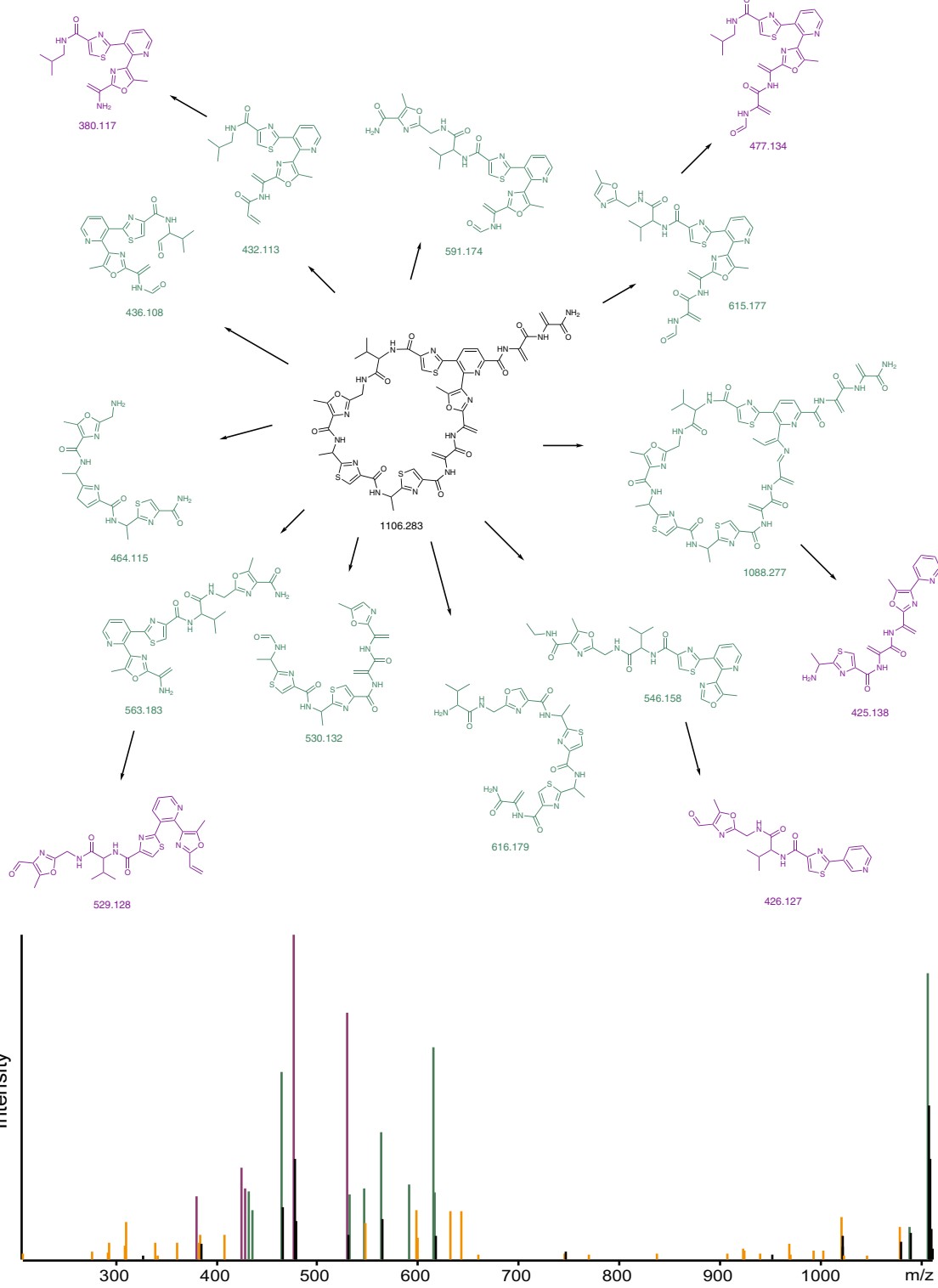

**Fig. 5 | Annotation of radamycin mass spectra from GNPS actinomycete dataset MSV000078937 using the Dereplicator+ model.** Neutral fragments are drawn for the 15 most intense explained peaks, with fragments at depth 1 drawn in green and fragments at depth 2 drawn in purple. Fragments are labeled with the mass-to-charge ratio of the peak that they explain. Arrows between fragments indicate that the target fragment is a sub-fragment of the source fragment according to the Dereplicator+ theoretical fragmentation model. At the bottom, peaks are colored if they were explained by the Dereplicator+ model and left as black otherwise. By switching from the string-based model [25] to this graph-based model, the score of the correct radamycin match increases from 9 to 49, and the p-value drops from $3 \times 10^{-17}$ to $3 \times 10^{-46}$. Dereplicator+ approximates p-values by assuming that theoretical fragment annotations occur as Bernoulli trials, as the description in Methods.

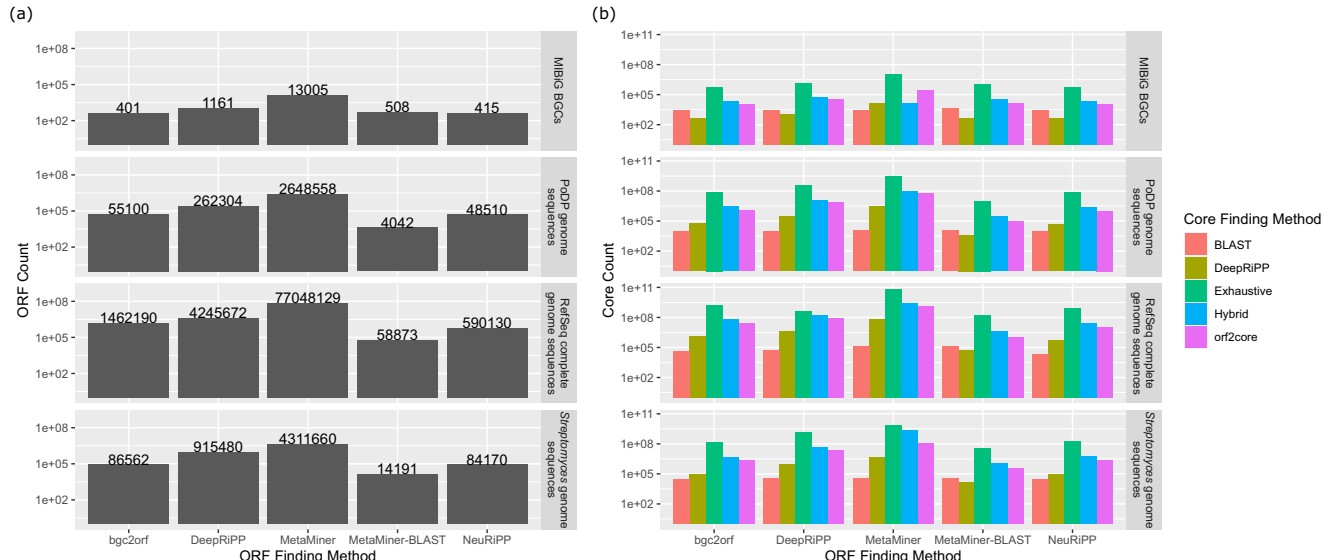

**Fig. 6 | A comparison of genome mining approaches on multiple datasets.**
PoDP, *Streptomyces*, RefSeq, and MIBiG datasets have 17,505, 48,542, 328,676 and 140 hypothetical BGCs, respectively. **a** The number of ORFs reported by each

method across the four datasets is illustrated. **b** The number of cores reported by each core detection method (represented by different colors) when analyzing ORFs found by each ORF detection method are listed on the bottom horizontal axis.

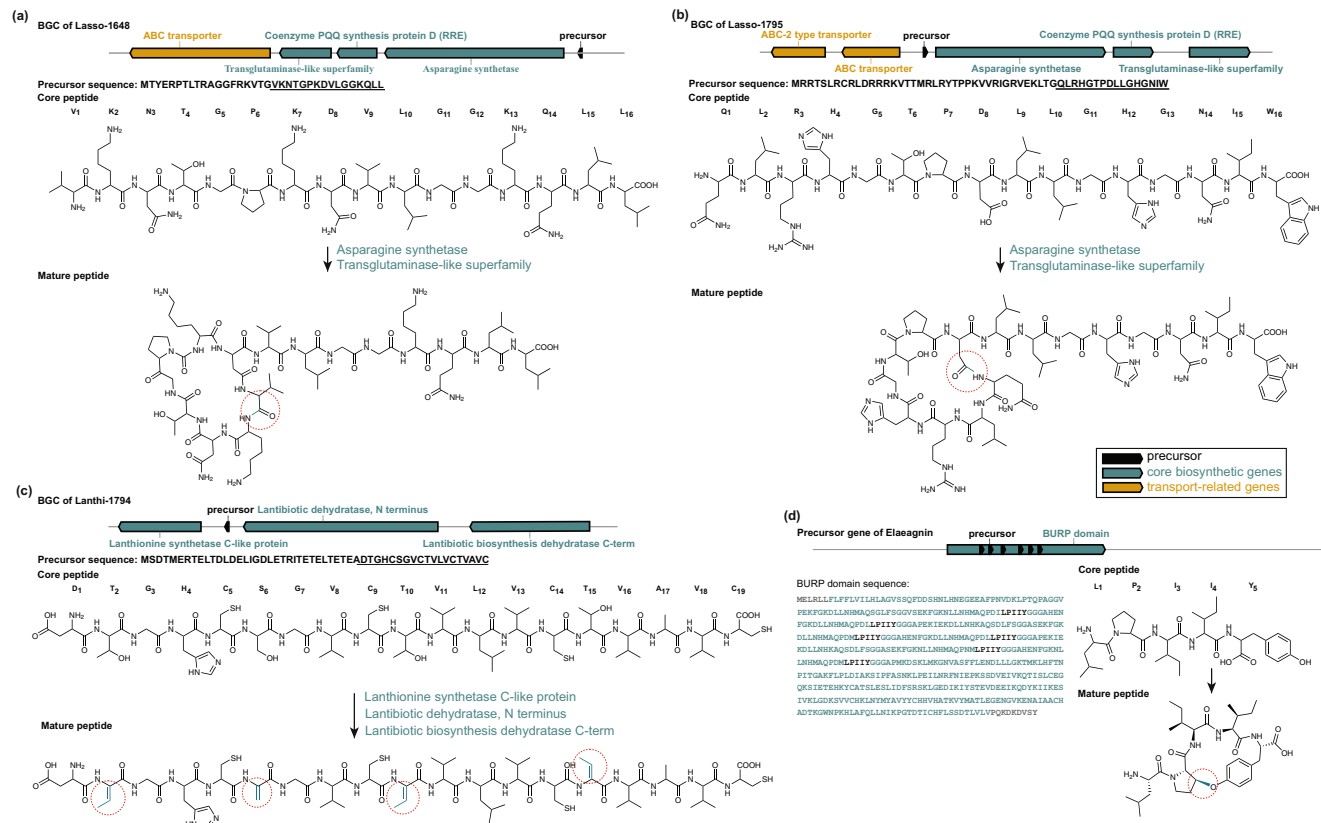

**Fig. 7 | RiPPs discovered with seq2ripp pipeline.** Two lassopeptides and a lanthipeptide discovered from *Streptomyces* (**a, b**) NRRL B-2660 and (**c**) WC-3904, respectively. In each BGC, the precursor gene is colored in black, core biosynthetic genes are colored in green, and the transport-related genes are colored in yellow. The width and location of the boxes demonstrate the relative length and position of each gene. Below each BGC, the precursor sequence is listed with the core sequence underscored. The structures of the core peptide, the mature peptide, and

the tailoring enzymes related to the PTMs are shown. PTMs are highlighted in green and circled in mature peptides. **d** A novel BURP-domain-derived cyclic pentapeptide was discovered from *Elaeagnus pungens*. BURP domain is highlighted in green and core sequences are colored in black. The chemical structure of elaeagnin is also shown at the bottom. Modification sites are emphasized by red dashed circles. MS/MS annotations for these three compounds are illustrated in Supplementary Figure 5–7. Structure elucidation of elaeagnin is shown in Supplementary Figure 20.

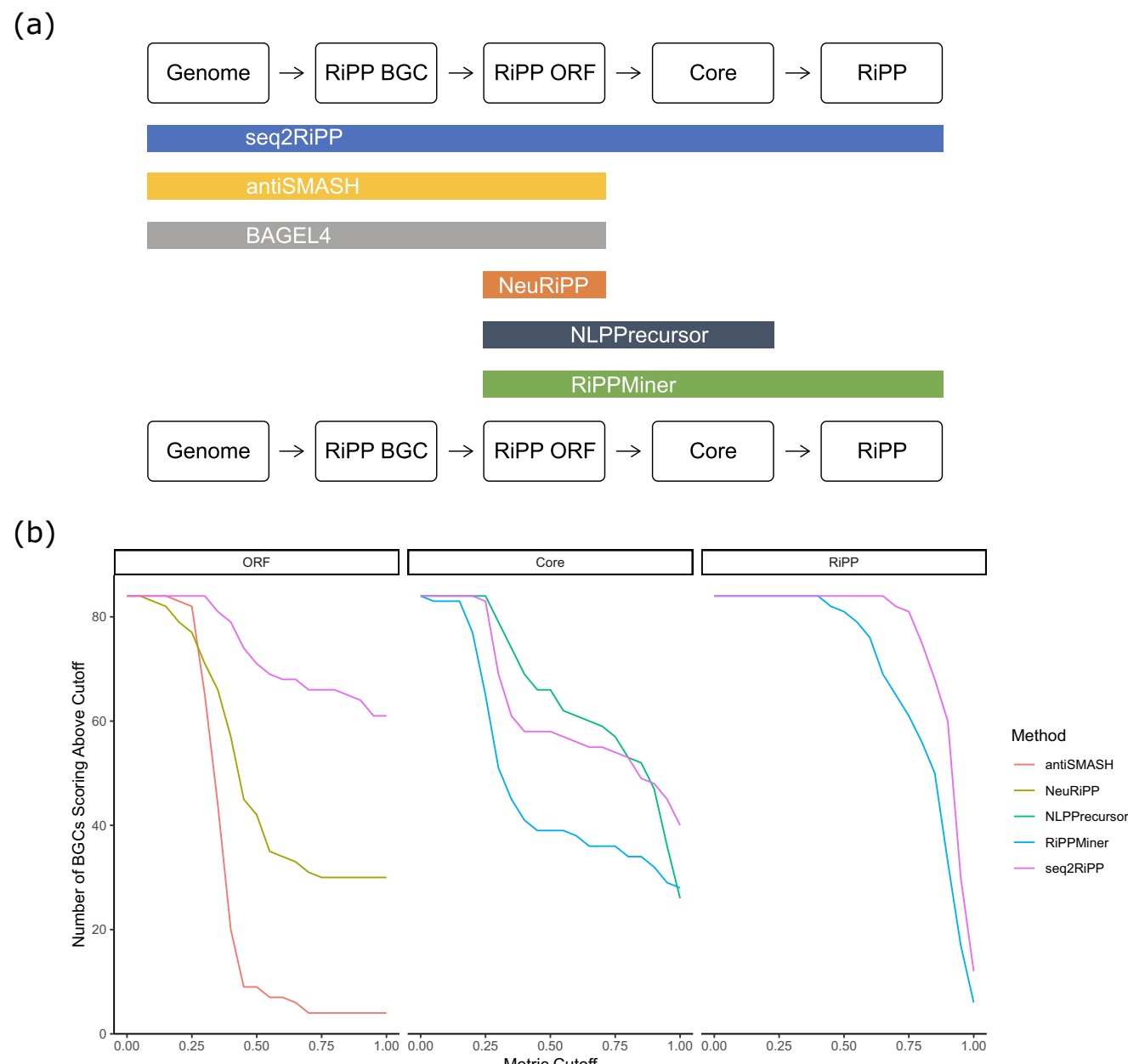

**Fig. 8 | Comparisons of seq2ripp pipeline and various RiPPs searching tools. a** A comparison of inputs and outputs for six benchmarked genome mining approaches for RiPP discovery. NLPPrecursor is a module of DeepRiPP. **b** A comparison of various tools for mining RiPPs from genomes. Metrics for ORF and Core detection are maximum normalized Levenshtein similarity to the correct core. For RiPPs, the metric was Tanimoto similarity between the true and predicted structures. Plots show the number of reference BGCs for which the tools' best predictions exceeded the metric cutoff.

peptides from an input list of putative precursor sequences using a deep neural network. DeepRiPP[25] is a tool with three modules: NLPPrecursor, BARLEY, and CLAMS. BARLEY failed to run in batch mode and CLAMS requires mass spectral input and therefore was not included in our genome mining benchmark. NLPPrecursor, is a module of DeepRiPP[25] that was included in this benchmark and uses a deep neural network to verify if an input sequence is a RiPP precursor peptide and, if so, it predicts the RiPP class and core sequence. The inputs and outputs for these tools are summarized in Fig. 8a.

The following tools, though relevant to RiPP discovery, were not benchmarked against. ThioFinder[52] cannot be run in batch mode and only supports thiopeptide BGC prediction. PRISM 4[24] is also unable to be run in batch mode. RiPPER[22] takes protein accessions as input and consequently is not suitable for a genome mining approach. The

publicly available version of RODEO2[53] is deliberately restricted to running on GenBank accessions, so it was not compared against. Pep2Path[54], HSEE[28], and CycloNovo[55] require mass spectral input to generate hypothetical RiPPs, while this benchmark experiment focuses only on genome mining approaches.

For each MIBiG RiPP BGC where a core and a molecular structure is reported in the literature, we downloaded the genome sequence of their producer microorganisms. In cases where genome sequences were not available, we used genome sequences of other strains within the same species with BLAST similarity of at least 96% against the ORF sequence with exact match to core sequence. Finally, 71 genome sequences encoding for 84 RiPPs were retained (Supplementary Table 1), and used as inputs for seq2ripp, BAGEL4[49], and antiSMASH6[50]. For tools that take ORFs as input (RiPPMiner[27], NeuRiPP[26], and

**Table 1 | Success rate of different genome mining approaches across 84 RiPPs**

| Median (Exact Matches) | | | |
|---|---|---|---|
| | ORF | Core | RiPP |
| seq2ripp | 1.0 (61) | 0.970 (40) | 0.932 (12) |
| antiSMASH | 0.353 (4) | | |
| BAGEL4 | 0.975 (42) | | |
| NeuRiPP | 0.487 (30) | | |
| NLPPrecursor | | 0.934 (26) | |
| RiPPMiner | | 0.375 (28) | 0.867 (6) |

Reported are the median metric and the number of exact matches of each method for each pipeline step it supported. ORF and core detection metrics are maximum normalized Levenshtein similarity to the correct core. The RiPP metric is the maximum Tanimoto similarity between true and predicted structures. The distributions of scores are shown in Figure S14.

NLPPrecursor[25]), seq2ripp ORFs were supplied as inputs for a comparison against seq2ripp's downstream pipeline.

We evaluated three metrics for the benchmarked tools: one for each predicted ORF, core, and RiPP. We assessed the accuracy of the ORF and core sequence predictions based on Levenshtein distance, which measures the minimum number of insertion, deletion, and substitution operations needed to convert one string into another. For ease of comparison, we further normalized the Levenshtein distance by the length of the reference sequence and converted it into a similarity score by subtracting the normalized distance from 1. Our ORF detection metric is the maximum normalized Levenshtein similarity between the correct core and the substrings of all of the predicted ORFs. Our core detection metric is the maximum normalized Levenshtein similarity between the correct core and the predicted cores. Our RiPP detection metric is the maximum Tanimoto similarity with radius 2 Morgan fingerprints[56] between the correct and the predicted structures. Tanimoto similarity between two molecules is the number of shared bits in their fingerprints divided by the total number of unique enabled bits across both fingerprints. All three metrics range between 0 and 1 with 1 denoting a perfect match. Seq2ripp outperformed all the other methods in the detection of ORFs, cores, and RiPPs (Fig. 8b, Supplementary Figure 13). Among 84 RiPP BGCs, seq2ripp correctly predicted 61 ORFs, 40 cores, and 12 RiPP molecular structures (Table 1, Supplementary Figure 14).

Two RiPPs, **lasso-1648** and **lasso-1795**, were discovered from *Streptomyces* sp. NRRL B-2660 datasets and one RiPP, **lanthi-1794**, was discovered from *Streptomyces* sp. NRRL WC-3904 datasets. To evaluate the performance of existing genome mining tools in predicting the RiPPs, these genomes were analyzed by antiSMASH6[50] and BAGEL4[49], and the ORFs containing the correct core for the RiPPs were used as inputs for RiPPMiner[27] and NLPPrecursor[25]. Similarity was measured using normalized Levenshtein similarity.

ORFs predicted by antiSMASH6[50] did not contain the correct core for any of the RiPPs. BAGEL4[49] did not predict any ORFs corresponding to **lasso-1648** and **lasso-1795** and in the case of **lanthi-1794**, the predicted cores were wrong.

Given ORFs containing the correct cores for the RiPPs from their genome assemblies, RiPPMiner[27] did not predict any cores and was therefore unable to predict the chemical structures of the RiPPs. Given the same input ORFs, NLPPrecursor[25] was able to correctly predict the core from **lasso-1648**, but failed to correctly predict the cores for **lanthi-1794** and **lasso-1795**.

These results indicate that a combination of other computational tools (without using seq2ripp modules) would not discover all of the novel RiPPs. The closest competitor module was NLPPrecursor, which was only able to retrieve one of the three cores.

## Discovery of bioactive ribosomal peptides from the human microbiota

Three N-formylated ribosomal peptides, fMEIVTSIISIIKTILG, fMAG-DIISTIVDFIKL, and fMNDLFGFITKVIDFLRSILVNGEPRR, were discovered by HypoRiPPAtlas from the human microbiota (MSV000080673) and subsequently tested for their bioactivity against human GPCRs[57].

Our results show that these peptides have significant agonist activity at formyl peptide receptor 1 (FPR1) by measuring the induction of β-arrestin 2 recruitment[57] (Supplementary Figure 15). These results demonstrate the potential of HypoRiPPAtlas in discovering bioactive ribosomal peptides.

## Discovery of cyclic plant RiPPs with novel PTMs

Plant-seq2ripp is a variant of seq2ripp algorithm tuned for discovering BURP-domain-derived RiPPs from plant species, i.e. lyciumin, legumenin, mono- and bicyclic cyclopeptide alkaloids, cercic acid, and stephanotic acid (Supplementary Figure 16)[35,36].

The algorithm was applied to transcriptomic and metabolomic datasets collected on 62 plant species (Supplementary Table 2) in order to test its potential for characterizing new RiPP chemistry from plants. Within these datasets, plant-seq2ripp correctly connected known plant RiPPs of known classes with corresponding BURP-domain precursor peptides, for example, legumenin with AhyBURP, lyciumin B with LbaLycA, cercic acid with CcaBURP1, stephanotic acid-[LV] with CcaBURP2, and cyclopeptide alkaloids selanine A and B with SkrBURP[36]. Moreover, the algorithm discovered plant RiPPs of known classes including four bicyclopeptide alkaloids (cores: ILLYPSY, VLFYRSY, FLLYPY, FLLYPSY) and one monocyclic cyclopeptide alkaloid (core: ILLY) derived from *Selaginella kraussiana* precursor SkrBURP, two lyciumins (core: QPFGVFAW, QPFGVFSW) derived from a BURP-domain precursor of *Jeffersonia diphylla*, and a stephanotic acid (core: QLKVW) derived from *Cercis canadensis* precursor CcaBURP2 (Supplementary Figure 17). Among these, *Jeffersonia diphylla* is a member of the Berberidaceae, and in this plant family no lyciumins have previously been reported. The validation of aforementioned plant RiPPs is generally conducted by transient expression of the matched BURP-domain precursor gene in *N. benthamiana* (Supplementary Table 3). For six of the plant RiPPs, we conducted transient expression of the matched BURP-domain precursor gene in *N. benthamiana* via *Agrobacterium tumefaciens* LBA4404 infiltration with the pEAQ-HT expression system[58]. After collecting LC-MS/MS data of methanolic extracts of the transgenic tobacco leaves six days after infiltration, MS/MS spectra corresponding to the observed BURP-domain ORFs were detected by seq2ripp, confirming that the predicted spectra are indeed derived from the predicted ORFs (Supplementary Figure 18).

Finally, plant-seq2ripp predicted eight plant RiPPs with new PTMs which included a BURP-domain-derived cyclic pentapeptide with a mass of 615.773 Da, a core peptide of LPIIY, and a Pro-Tyr-macrocyclization from *Elaeagnus pungens* (Fig. 7 and Supplementary Figure 19). This macrocyclization was not a part of the seq2ripp PTM database but using variable mass spectral search[59] seq2ripp was able to discover this PTM without any a priori information. This predicted crosslink was characterized by NMR of the purified natural product (Supplementary Figure 20 and Supplementary Table 4) named elaeagnin. The elaeagnin PTM, a macrocyclic ether bond between the β-carbon of a proline and the phenol-hydroxyl-group of a tyrosine, was previously proposed in a plant peptide derived from soybean BURP-domain precursor GLYMA_04G180400[36] (Supplementary Figure 21). Elaeagnin features an unmodified N-terminus in contrast to N-termini of characterized BURP-domain-derived RiPPs which are modified as a glutamine-derived pyroglutamate or via N,N-dimethylation[36]. Elaeagnin structure elucidation by 1D and 2D NMR and Marfey's analysis (Supplementary Figure 20 and Supplementary Table 4) establishes the

structure of this RiPP class and it was verified as a RiPP by transient expression of the corresponding BURP-domain precursor gene derived from the *E. pungens* transcriptome as a truncated precursor with five core peptide repeats, in *N. benthamiana* and subsequent LC-MS/MS-based detection of elaeagnin in methanolic extract of transgenic tobacco leaves after six days (Supplementary Figures 18 and 22). The discovery of elaeagnin showcases the power of seq2ripp in identifying classes of RiPPs with novel modifications.

### HypoRiPPAtlas server

The HypoRiPPAtlas web server currently includes hypothetical RiPPs from 22,671 complete microbial genomes from RefSeq. Users can select organisms and download the corresponding BGC, ORF, core, and molecular structure data. Additionally, the HypoRiPPAtlas web server supports the processing of inputpaired genomic and spectral data from users.

## Discussion

Breakthroughs in genome mining and mass spectrometry data collection have revolutionized the field of natural product discovery during the last decade. Development of popular genome mining tools such as antiSMASH has made it possible to quickly profile microbial genomes for detecting natural product BGCs. However, the current state-of-the-art approach for connecting BGCs to molecules is through the expression of the BGC in a heterologous host and subsequent isolation and structure elucidation of the product, which is a slow and expensive process. Therefore 99% of BGCs extracted from microbial genomes and stored in public repositories remain orphan, i.e. they are not linked to any small molecules.

On the mass spectrometry front, the development of the GNPS repository along with molecular networking has made it possible to annotate known natural products and discover their novel variants. Currently, GNPS hosts more than a billion mass spectra from more than five hundred laboratories. However, only 2% of GNPS spectra have been annotated as known molecules or their analogs. It has been hypothesized that a portion of the annotated spectra from GNPS is likely to correspond to orphan BGCs from microbial genomes.

To fully utilize the power of recently developed repositories of microbial BGC and mass spectral datasets, computational techniques for high-throughput linking of BGCs to mass spectra are needed. However, in order to link mass spectral datasets to BGCs from microbial genomes, one first needs to predict the hypothetical structure of the molecular product of these BGCs. In order to fill in this gap, we present HypoRiPPAtlas, a public repository of hypothetical natural products predicted by mining microbial genomes. In the case of RiPP natural products, using the seq2ripp algorithm we populated this Atlas with hypothetical RiPPs from 22,671 complete microbial genomes available from RefSeq.

Seq2ripp identified three microbial RiPPs from the PoDP datasets, and ten plant RiPPs from 62 plant metabolomic and transcriptomic datasets. Through variable mass spectrometry, seq2ripp discovered a plant RiPP with a novel PTM, which was not included in the PTM training set of the algorithm, from Elaeagnaceae, highlighting the power of this method for the identification of novel classes of natural products that were missed by the previous approaches. Bgc2orf and orf2core modules within seq2ripp are capable of identifying RiPP precursors and cores without overfitting.

Hypothetical molecules from HypoRiPPAtlas can be filtered to user-specified taxonomic clades, and the retained molecules can be queried against mass spectral datasets using Dereplicator+, an in silico database search tool available from GNPS. Previous techniques modeled RiPPs as strings of amino acids, along with post-translational modifications. In contrast, HypoRiPPAtlas models RiPPs as graphs with atoms and chemical bonds, improving accuracy in representations of post-translational modifications (e.g. cyclizations). The graph model

also improves the accuracy of in silico methods in predicting the fragmentation of RiPPs containing nonstandard amino acids (e.g. oxazole and thiazole).

Another challenge in linking mass spectral datasets to microbial BGCs is that over 99% of spectra from GNPS are collected on strains with unknown DNA sequences from complex environments. HypoRiPPAtlas infrastructure supports searching mass spectral datasets against taxonomic clades, allowing for natural product discovery from datasets without genomic information. This often results in the discovery of natural products from mass spectra of one strain against the genome of a different strain. For example, radamycin was identified by searching mass spectra of a marine *Streptomyces* against the genome of a tomato flower symbiont *Streptomyces*.

Currently, HypoRiPPAtlas reports on average 84 hypothetical ORFs, 1605 hypothetical cores, and 1753 hypothetical molecules per RiPP BGC. Multiple RiPPs have been recorded per BGC because (i) many RiPP BGCs have multiple molecular products, e.g. many cyanobactin and plant RiPPs have multiple cores per each ORF, and (ii) the activity of many RiPP enzymes remain ambiguous resulting in multiple possibilities. By keeping track of multiple hypothetical RiPPs per BGC, we can increase the chance of capturing all the correct RiPPs. In comparison to the state-of-the-art RiPP identification tools, seq2ripp captured more correct cores, ORFs, and RiPPs. However, the natural drawback of this strategy is that the majority of RiPPs in the Atlas will be spurious and require validation via mass spectrometry before downstream analysis. As our mass spectral search tools improve, prioritizing sensitivity over specificity in our genome mining tools becomes more appealing. By searching against mass spectral repositories, HypoRiPPAtlas enables the identification of a large number of RiPPs, providing the training data for machine learning approaches to improve the prediction accuracy of post-translational modifications.

HypoRiPPAtlas currently only supports ribosomally synthesized and post-translationally modified peptides. For a more comprehensive atlas, we are currently working on extending functionality to support non-ribosomal peptides and polyketides.

## Methods

### Natural product discovery by HypoRiPPAtlas

The pipeline for natural products discovery by HypoRiPPAtlas consists of the following steps.

**Extracting BGCs and predicting hypothetical RiPPs**. BGCs are either imported from IMG-ABC, antiSMASH-DB, and BiG-SLiCE, or mined from RefSeq/IMG-M. Hypothetical RiPPs are predicted from BGCs using seq2ripp.

**Filtering the Atlas using taxonomic information**. The entire Atlas is too large for downstream analysis. Users can filter the Atlas using specific taxa terms, and then download the entire hypothetical molecules in that clade in the SMILES format, along with corresponding BGCs, ORFs and cores. For example, the Atlas contains 20 BGCs, 48 ORFs, 273 cores and 120,701 molecules for the strain *Streptomyces globisporus* NRRL B-2709.

**Predicting spectra of hypothetical molecules from the Atlas**. We used Dereplicator+ to pre-calculate the fragmentation graphs for all the molecules from the Atlas. The fragmentation graphs are stored as binary files, and they are also downloadable for user-specified taxonomic clades. It is much faster to search mass spectra against precalculated fragmentation graphs using Dereplicator+ than it is to search against raw SMILES structures.

**Database search of mass spectra against HypoRiPPAtlas**. Mass spectral datasets can be searched against the SMILES structures/precalculated fragmentation graphs using Dereplicator+ from the GNPS

infrastructure. HypoRiPPAtlas has been designed in a way that its interface is fully compatible with GNPS.

**Reporting false discovery rates.** False discovery rate (FDR) based on target decoy analysis reported by Dereplicator+ are used.

**Computing statistical significance of PSMs.** Dereplicator+ approximates p-values by assuming that theoretical fragment annotations occur as Bernoulli trials. The whole fragmentation graph is assigned a probability of achieving a score at least as high as the observed score $n$ by computing the probability that a Poisson-Binomial distribution consisting of Bernoulli trials for each theoretical fragment has at least $n$ successes. The approximation process is described in Dereplicator+[29].

**Molecular networking.** Currently, feature-based molecular networking[15] supports annotations from Dereplicator+, and these networks can be visualized through the GNPS infrastructure.

## Overview of MetaMiner

In the past, we developed MetaMiner, a computational technique for discovery of RiPPs[19]. The MetaMiner pipeline analyzes the paired genome/metagenome assemblies and tandem mass spectra from isolated microbes or bacterial or fungal communities. Starting from the genome assemblies, MetaMiner (i) identifies putative BGCs and their corresponding precursor peptides, (ii) constructs target and decoy putative RiPP structure databases modeling them as strings, and (iii) matches tandem mass spectra against the constructed RiPP strings using Dereplicator[60].

## Overview of seq2ripp algorithm

Given a microbial genome sequence in fasta format, seq2ripp predicts hypothetical RiPP molecules (in the SMILES format) in the following steps.

**From genome to BGCs.** For the identification of RiPP enzymes in the genome sequences, we used hidden Markov models (HMMs), which are popular models extensively used for the identification of protein motifs in genome sequences[18]. We collected 26 HMM profiles from Table S3 in Metaminer[19]. An additional 126 HMM profiles were manually collected by literature search and from Dataset S6 in DeepRiPP[25]. They were either collected with their Pfam ID or generated with hmmbuild from gene sequences in literature. Relevant references were taken from all RiPPs in the MIBiG[47] repository. In total, 152 HMM profiles (Supplementary Table 5) are used by genome2bgc for the identification of RiPP BGCs and the detection of tailoring enzymes. Hits to HMM profiles are filtered by an HMMER E-value threshold of $10^{-5}$. Domains of unknown function (DUFs) are not automatically detected by the genome2bgc module. However, we do include DUFs that have been previously linked to the biosynthesis of RiPPs, such as DUF4135 for the synthesis of lanthipeptides[61].

**From BGCs to ORFs.** ORFs are extracted from BGCs by bgc2orf, a deep neural network (Fig. 3a). Additionally, we also support the exhaustive strategy recruited by MetaMiner[19], a BLAST-based strategy, and strategies from DeepRiPP[25] and NeuRiPP[26]. In the exhaustive strategy, we consider all short ORFs with lengths longer than 10 aa as feasible ORFs. While this strategy is very sensitive, it can result in a high number of false positives. In BLAST-based strategy, ORFs identified by the exhaustive search strategy are aligned against a database of 525 known RiPP ORFs with blastp, and those with E-value lower than 0.01 are retained. In bgc2orf, we train a sequence classification model which contains a Convolutional Neural Network (CNN) and a Long Short-Term Memory (LSTM) Network, based on 2,726 amino acid sequences of known RiPP ORFs and 19,224 amino acid sequences of non-RiPP ORFs[26]. Then for each short ORF in the BGC, we will predict whether

the ORF is a structural ORF or not, and only consider those with high probabilities (higher than 50%) as hypothetical structural ORFs.

Bgc2orf is trained as follows: first, all input sequences are padded to a length of 200 amino acids. Each amino acid, including the padding symbol, is tokenized and embedded into a vector of size 100. The model includes two 1D CNNs. One CNN convolves the input sequence in terms of its topology, the other CNN convolves the tokenized vectors. The convolution on the sequence helps the model to detect the RiPP features on amino acids; the convolution on the token vector summarizes the embedded character information in high dimensional space. The outputs from the two CNNs and the input embeddings are concatenated and fed into a single-layer bidirectional LSTM. The LSTM learns and summarizes the sequential features from the amino acid chain. The output of the LSTM is flattened and converted to a binary output with a flattened layer and a dense layer. The prediction loss is calculated by cross-entropy loss during the training of the model. The learning rate begins with 1e-3 and decreases 10% every 40 epochs.

**From ORFs to cores.** Core peptides are detected from ORFs by orf2-core, a deep neural network (Fig. 3b). Similar to the previous step, we also support an exhaustive strategy from MetaMiner[19], DeepRiPP[25], and a BLAST-based strategy. In the exhaustive strategy, all the peptide fragments with lengths between 3 to 30 aa are considered as candidate core peptides. In the BLAST-based strategy, the ORFs are aligned with 525 known RiPP cores by blastp, and the part of ORFs aligning with the core sequence with e-value lower than 0.001 are extracted (allowing for an error of up to two bases on each side). In orf2core, we framed the task as a phoneme discovery problem, where the input is an amino acid sequence, and the output is putative cleavage sites. We then trained a CNN-LSTM-Conditional Random Field (CRF) model on the cleavage site information of 3169 known RiPP ORFs, detailed below.

We use a discriminatory deep learning model to predict core and non-core frames of sequences given the amino acid sequence of an ORF as the continuous input. All input sequences are padded to a length of 200 amino acids. Each amino acid is tokenized and embedded into a vector of size 25. The model includes two 1D CNNs. One CNN convolves the input sequence in terms of its topology, and the other CNN convolves the tokenized vectors. The convolution on the sequence helps the model detect amino acids surrounding the enzymatic cleavage site. The convolution on the token vector summarizes the embedded character information in high-dimensional space. The outputs from the two CNNs, and the input embeddings are concatenated and fed into a single-layer bidirectional LSTM. The LSTM learns the translation from the amino acid chain to core and non-core frames of sequences. The prediction loss is calculated by a conditional random field layer, which calculates the negative log-likelihood during the training of the model, and performs the Viterbi algorithm to optimize labels during prediction. An additional approach will be triggered when repeat patterns are observed in an ORF, by searching repeated leader and follower patterns in the sequence (Fig. 3c, Supplementary Figure 23).

**From cores to RiPPs.** In order to use Dereplicator+ for the discovery of RiPPs, one needs to derive the complete chemical structure graph of RiPPs from their BGC (rather than a precursor peptide along with modification masses, required by Dereplicator). To enable this, we start with precursor peptides and model tailoring enzymes as graph-modifiers. Each enzyme searches a particular chemical motif in the molecular graph of a RiPP, and whenever it finds the motif, it optionally applies the corresponding tailoring modification (Fig. 9).

Core2ripp predicts hypothetical molecules by applying modifications corresponding to tailoring enzymes present in the BGC. We do this by extracting all the information from the known RiPP tailoring enzymes and their corresponding modifications by literature mining[4], and parsing them in a computer-readable format. Our format consists of a motif (stored as a SMILES string) along with a series of graph

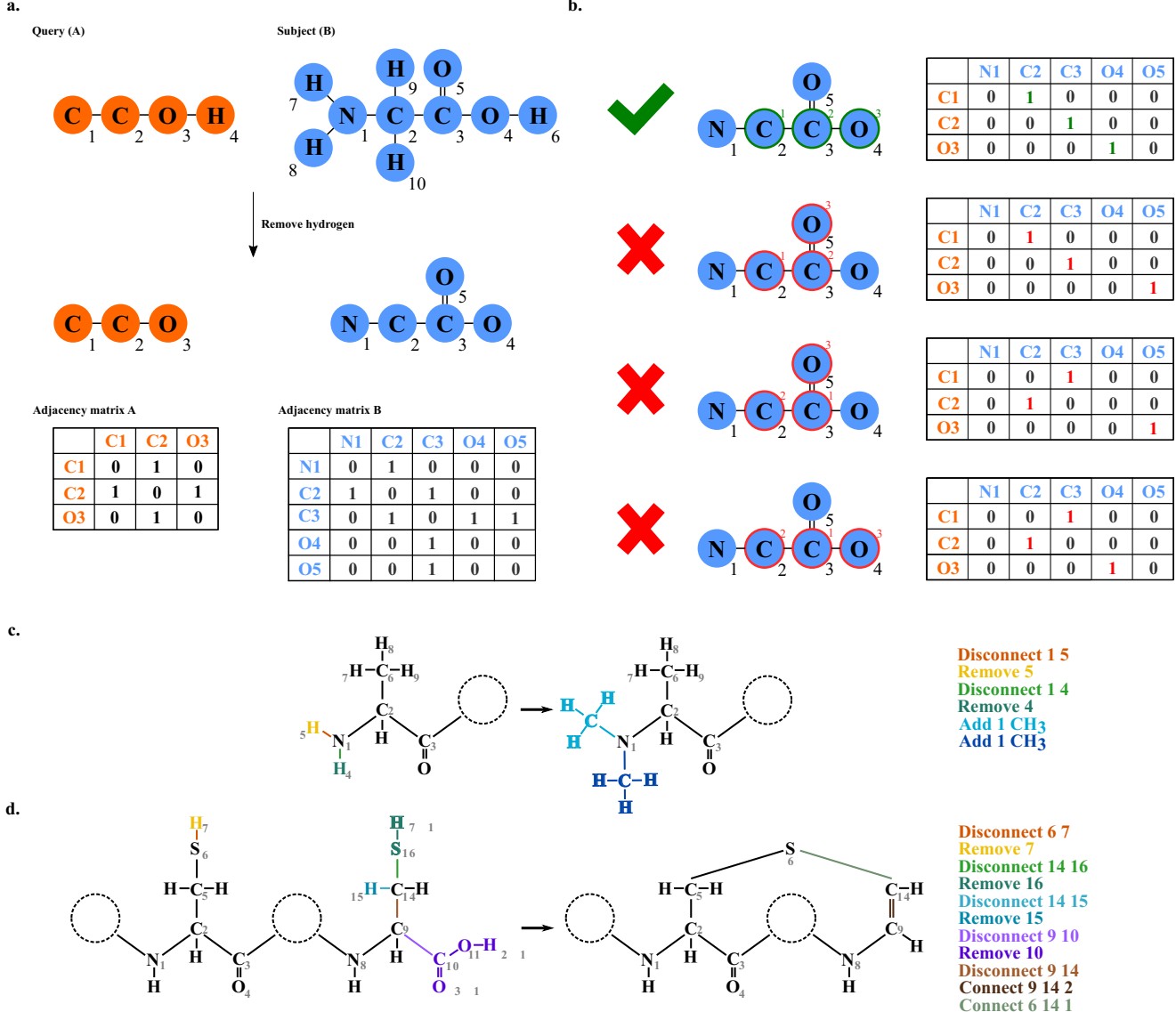

**Fig. 9 | An illustration of finding motifs through subgraph isomorphism and motif-based modification.** In (**a**) and (**b**), the query is highlighted in orange, while the subject is highlighted in blue. **a** First, the hydrogens are removed from the query and the subject. Next, the atoms are labeled by a breadth-first traversal of each graph from the first non-hydrogen atoms in the input file. Finally, an adjacency matrix for each graph is prepared. (**b**) Only the mappings that define an isomorphism are kept. Converting known enzymatic modifications of RiPPs into a computer-readable format for (**c**) dimethylation of N-terminal, e.g. in cypemycin[76] and (**d**) oxidative decarboxylation, e.g. cypemycin and epidermin[77]. In our format, we use commands disconnect/connect (for bonds) and add/remove (for chemical substructures). For example, in part (**c**), "disconnect 1 5" removes the bond between the nitrogen atom with index 1 and the hydrogen atom with index 5, while "remove 5" removes the hydrogen atom with index 5, and "Add 1 $CH_3$" adds a methyl group to the nitrogen atom indexed 1 (methylation). In part (**d**), "connect 9 14 2" adds a double bond between atom 9 and atom 14. Each step of the modifications is differentiated by different colors, both in the action on the right and in the molecule on the left. We converted all the modifications from Arnison et al.[4], Montalbán-López et al.[78], and our in-house database of modifications into this format (Supplementary Figure 24 and Supplementary Table 6).

modifications (addition/removal of nodes and edges) that are applied to the molecular structure whenever the motif is observed (Fig. 9). Supplementary Figure 24 and Supplementary Table 6 summarize all the modifications from Arnison et al. (2013)[4], and our in-house database[19]. Core2ripp does not contain any learned parameters, instead relying on enzymatic domain-modification associations extracted from literature.

Core2ripp models RiPPs as graphs, where nodes represent atoms and edges represent chemical bonds. This graph representation not only facilitates handling complex post-translational modifications, but also enables saving hypothetical RiPP structures in molecular formats such as SMILES, which are commonly employed by small molecule mass spectral database search tools such as Dereplicator+ and

MolDiscovery[29,62]. This graph-based database search scoring improves on the string-based scoring of MetaMiner in that (i) Dereplicator+ handles fragmentations of N-C, O-C and C-C bonds (rather than amide bonds only), improving its accuracy in modeling fragmentation of peptides containing nonstandard amino acids, e.g. dehydrated alanine, and dehydrated butyrine, oxazole and thiazole, and (ii) Dereplicator+ utilize a fragmentation graph model that enables capturing higher depth fragmentations (fragmentation depth is the number of times that a graph can be fragmented recursively).

Given a core sequence and a list of tailoring enzymes, core2RiPP predicts all the hypothetical RiPP structures using the following steps. First, the chemical structure of the core sequence is represented as a graph, where each vertex is an atom with an index number and the type

of atom, and each edge is a bond between two indexed atoms and the type of chemical bonds. Next, for each modification, the locations of motifs are collected by subgraph isomorphism[63] (see details below). Finally, the putative combinations of modifications will be calculated and applied to the core sequence (see details below). The final product will be stored in the SMILES format.

**Finding motifs by subgraph isomorphism.** We model the problem of searching motifs in the precursor RiPP sequence as a subgraph isomorphism problem. In the subgraph isomorphism problem, given graphs A (query) and B (subject), the problem is to determine if there is a subgraph of graph B that is isomorphic to graph A. Here, the subject is the chemical structure of the precursor peptide sequences, and the query is the chemical structure of the motifs. For both the subject and the query, vertices are atoms, and the edges are bonds. The subgraph isomorphism problem is shown to be NP-hard[64].

We slightly revise Ullman's subgraph isomorphism algorithm for motif finding[63], which makes it three orders of magnitude faster than the original approach. Ullman's subgraph isomorphism algorithm builds an $m$ by $n$ binary correspondence matrix, where $m$ is the number of nodes in the query, and $n$ is the number of nodes in the subject. The correspondence matrix has a 1 whenever the following two constraints hold: (i) the corresponding query and subject atoms have the same label (e.g. they are both carbon), and (ii) the multiset of neighboring nodes for the query is a subset of the multiset of neighboring nodes for the subject. Then, for every row in the correspondence matrix, a single column containing a 1 is selected. This results in a mapping from query node indices to subject node indices. Then, the algorithm checks whether this mapping defines an isomorphism, i.e. all the edges in the query are present in the subject. Whenever this constraint is violated, the algorithm backtracks to select a new column with 1 from one of the rows.

We modified the Ullman algorithm by incorporating edge-level labels: the multiset of neighboring node and edge pairs for the query is a subset of the multiset for the subject, where the edge is labeled as a single/double/triple bond. We further remove the hydrogen atoms from both query and subject to accelerate the algorithm and avoid unnecessary computations. Moreover, by constructing the query and the subject into spanning trees, we are able to enforce that in each iteration, the atom selected from the query/subject be connected to the query/subject atoms added in the previous iterations without additional computations (Fig. 9a, b).

**Motif-based modification.** MetaMiner models RiPPs as strings of amino acids, and then modifications are applied to amino acids as mass shifts. In contrast, seq2ripp models RiPPs as graphs, and modifications are applied as graph modifications on the reaction motifs. We define modifications in a computer-readable format, using four actions: add, remove, connect, and disconnect. The add/remove actions are used for adding and removing atoms, while connect/disconnect actions are for adding and removing edges (Fig. 9c, d).

Given a list of tailoring enzymes in the BGC, core2ripp predicts the hypothetical products by adding the different combinations of modifications on the core peptide according to Supplementary Figure 24 and Supplementary Table 6. If there are 10 feasible modifications at different locations, this procedure produces 1024 possible products. These hypothetical structures are saved in SMILES format.

**Construction of training/validation/test datasets and cross-validation**
For training bgc2orf, We collected a dataset of 2726 known RiPP (positives) and 19224 ORFs that are not RiPPs (negatives)[26]. We allocated 81% of these data-points for training, 9% for validation, and 10% for testing. To avoid data imbalance, negative data is sampled randomly to match the number of positive training/validation/testing

datasets. The validation datasets are used to evaluate models during the training process but not to update the model weights. To avoid leakage, the test data is only used for the final evaluation after models are trained.

For orf2core, we used a dataset of 3169 known ORFs with their cores. We allocated 81% of these data-points for training, 9% for validation, and 10% for testing. The validation datasets are used to evaluate models during the training process but not to update the model weights. To avoid leakage, the test data is only used for the final evaluation after the models are trained.

**Orf2core architecture design and hyperparameter tuning**
In orf2core, we frame the task of identifying cores from ORFs as a sequential pattern-searching problem. Orf2core is designed to capture two correlated objectives simultaneously, (i) the peptidase cleavage sites, and (ii) the class-specific pattern of the core sequence. Therefore, the model learns to predict the core/non-core sequences based on these two objectives.

Core peptides are cleaved by peptidases. Peptidases usually have high substrate specificity based on their local biochemical and biophysical properties, such as the composition of amino acids in the peptide sequence, the binding affinity of surrounding amino acids to the peptidase, and the accessibility of the site to the peptidase in three-dimensional space. Predicting the cleavage site of peptidases is a challenging problem[65].

Additionally, it is possible to infer the class of peptidases based on the class of RiPPs, which in turn can be predicted from the amino acid sequence of ORFs and other enzymes present in the BGC. For example, the peptidases that cleave class I lasso peptides are usually very different from those that cleave class II peptides, with distinct cleavage sites.

Because of variations in the length of input peptide sequences, the inputs are either padded with a placeholder or truncated to a length of 200 to ensure the data is represented by vectors of fixed size. Additionally, to convert a string of amino acids into numerical values, an embedding layer is necessary. Each amino acid is embedded to a vector of size 100.

Since the input peptide sequence is a string of amino acids, and the model needs to predict the core versus non-core label of each amino acid, we first incorporated a long short-term memory conditional random field (LSTM-CRF) layer, which is a popular machine learning model for incorporating string inputs for label prediction[66] (Supplementary Figure 25a, b). However, if only one layer of LSTM-CRF is used, the highest accuracy for prediction of the cleavage site is 54.20% before starting to overfit. Since the surrounding amino acids are informative for the prediction of the cleavage sites, a one-dimensional CNN layer is further added to incorporate the information from surrounding amino acids, increasing the accuracy from 54.20% to 62.59%[67] (Supplementary Figure 25c–g). Additionally, to avoid overfitting the model, regularization methods such as dropout and ReLU are added. These regularization methods prevent an increase of the validation loss and reduce its fluctuation during the training process[68,69]. Adding these regularizations increased the accuracy from 62.59% to 73.08%, without overfitting (Supplementary Figure 25h–o). The highest accuracy was achieved by embedding size of 100 (Supplementary Table 7a), strides of length 1 and kernel size of length 5 (Supplementary Table 7b), and dropout with $p = 0.2$ before CRF layer, along with a ReLU layer (Supplementary Table 7c). Training and validation loss plots show that the model is stable and not overfitting (Supplementary Figure 26). All of the experiments are conducted on a single GeForce RTX 2080 Super 8GB GDRR6.

**Bgc2orf architecture design and hyperparameter tuning**
In brg2orf, we frame the task of identifying RiPP ORF as a sequential pattern-searching problem. Bgc2orf is designed to recognize the

class-specific patterns in ORFs. The architecture of bgc2orf is identical to orf2core, except for the final layer, where the CRF layer is replaced with a linear layer. This layer outputs a probability that the input ORF is a RiPP ORF. Moreover, in order to avoid overfitting a ReLU layer is added after each 1D CNN layer. These changes increase the validation accuracy from 94.46% to 95.94% (a cut-off of 0.5 is used).

## Bioactivity of human microbiome peptides

The three peptides from human microbiota were synthesized by GenScript Biotech, New Jersey. Screening of compounds in the PRESTO-Tango GPCR-ome was accomplished using previously described methods with several modifications[57]. First, HTLA cells were plated in DMEM with 2% dialyzed FBS and 10 U/mL penicillin-streptomycin. Next, the cells were transfected using an in-plate PEI method[70]. PRESTO-Tango receptor DNAs were resuspended in Opti-MEM and hybridized with PEI prior to dilution and distribution into 384-well plates and subsequent addition to cells. After overnight incubation, drugs diluted in DMEM with 1% dialyzed FBS were added to cells without replacement of the medium. After another overnight incubation, the culture medium and peptide-including buffer were removed by aspiration. 20 $\mu$l of Bright-Glo solution (Promega) diluted 20-fold in assay buffer was added into each well. The cells were incubated with buffer at room temperature for 15-20 minutes. Then, the luminescence was measured in a Trilux luminescence counter. The relative luminescence units (RLU) were collected from the machine and calculated by fold change based on the basal RLU.

## Overview of plant-seq2ripp algorithm

Plant-seq2ripp is structured similiar to seq2ripp algorithms, with the differences that (i) plant-seq2ripp allows for transcriptomics input data in fasta format, instead of genomics, and (ii) a plant-specific set of enzymes and modifications are used. Transcriptomics data was assembled using SPAdes (v3.13 (version of the assembled transcriptomes))[71].

## The HypoRiPPAtlas server

HypoRiPPAtlas was built using Svelte and Kubernetes, fully hosted on Amazon Web Services. Data for BGCs, ORFs, cores, and RiPPs from publicly available data can be browsed. Users can also upload their own genomes and mass spectra to run seq2ripp. Results can be both visualized and downloaded. A walkthrough of usage of the server can be found at https://github.com/mohimanilab/seq2ripp.

## Data availability

All the MS datasets are publicly available from the GNPS infrastructure under the following accession code (microbes: Supplementary Table 8; plant: MassIVE MSV000088918 [ftp://massive.ucsd.edu/MSV000088918/ and https://gnps.ucsd.edu/ProteoSAFe/result.jsp?task=afd60fcca2194035849ef9bbe4657da2&view=advanced_view]. All plant transcriptomics datasets have been deposited and publicly available in the Zenodo[72] database under accession code 6253747 [https://zenodo.org/record/6253747]. The genome sequences of all the microbial organisms investigated in this study are available from NCBI RefSeq (GCF_000718455.1, GCF_000719185.1, GCF_000718755.1, GCF_000720725.1). The biosynthetic gene clusters analyzed in this study are available from MIBiG (BGC0001753). The gene sequence of BURP-domain precursor peptide EpuBURP has been deposited in GenBank (OR257605).

## Code availability

HypoRiPPAtlas is available from https://github.com/mohimanilab/seq2ripp[73]. Instructions for using HypoRiPPAtlas as a web-server are also available on the GitHub page.

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

## Acknowledgements

The work of Y.L., H.L., N.M., L.C., M.G., A.K., K.N. and H.M. was supported by a National Institutes of Health New Innovator Award DP2GM137413, a National Science Foundation Award DBI-2117640, and a U.S. Department of Energy award DE-SC0021340. This work used the Extreme Science and Engineering Discovery Environment (XSEDE), which is supported by National Science Foundation grant number ACI-1548562. Specifically, it used the Bridges system, which is supported by NSF award number ACI-1445606, at the Pittsburgh Supercomputing Center (PSC). The work of D.N.C. was supported by a PhRMA foundation predoctoral fellowship and the work of R.D.K. was supported by the UM Biosciences Scholar Program and a National Institute of Health award (R35GM146934). We also thank Prof. George Lomonossoff (John Innes Centre, UK) for sharing the pEAQ-HT vector. The work of Y. L. was done during his time at Carnegie Mellon University. We would like to thank Louis-Felix Nothias and Mert Inan for their insightful comments on the manuscript.

## Author contributions

Y.L., N.M., C.M., B.K., H.L., L.C., M.G., A.K and K.N. implemented seq2-ripp algorithm and performed the analysis. A.G. designed the interface of the Atlas with mass spectrometry data. D.N.C. and R.D.K. analyzed plant datasets and characterized plant and microbial peptides. S.S. and B.R. tested the activity of peptides from human microbiota. M.G. and S.W. designed and implemented the HypoRiPPAtlas server. R.D.K., B.B. and H.M. designed and directed the work. Y.L., R.D.K. and H.M. wrote the manuscript.

## Competing interests

H.M. and B.B. are co-founders and have equity interests from Chemia.ai, LLC. The other authors declare no competing interests.

## Additional information

**Peer review information** : *Nature Communications* thanks Tomáš Pluskal and the other, anonymous, reviewer(s) for their contribution to the peer review of this work. A peer review file is available.

