## [Peer Review File · Nature Communications]

REVIEWER COMMENTS

Reviewer #1 (Remarks to the Author):

The authors have carefully addressed all my comments from the previous round of review. I have no further comments or suggestions.

Reviewer #3 (Remarks to the Author):

This manuscript from the Mohimani lab describes HypoRiPPAtlas, which is an algorithm that links RiPP genome mining with the global natural products social molecular networking program. This is a novel and important concept, given that the latter houses a large amount of mass spectrometry data for natural products from various biological extracts. Based on a reading of the prior reviews, the authors have improved some aspects of the paper, but unfortunately, the paper is not ready for publication. Some rather egregious omissions and incorrect statements in the original manuscript appear to have been corrected, but some new text sections range from misleading to factually incorrect. Equally troubling is that the text itself is meandering, disorganized, and of unjustifiable length. The narrative is often confusing and some of the sections seem chronologically out of place. With respect to the figures, the presentation is below average. Based on several factors, most notably the large number of errors, the authors give the impression they do not know the RiPP field and haven't read the papers in their own citation list. A major rewrite is necessary before this manuscript should be published in any journal, let alone a multidisciplinary journal.

Page 3: To call GNPS a "gold mine" equates to hyperbole. In my experience, this database is not particularly well suited for RiPP discovery and the quality of the MS data is so highly variable that it often precludes usage.

Page 5, elsewhere: Why do the authors state in multiple locations that RODEO is only applicable to lasso peptides? Have the authors read the numerous papers (from their own citation list) that show RODEO being used for other RiPP classes? I have never heard anyone else say that RODEO only works for lasso peptides.

I don't follow the logic progression of including SynBNPs in this paragraph. This is a really big pivot. One idea per paragraph, please. The same is true throughout the paper. This is a criticism to help the authors produce a more readable text.

"non-class-specific"? Class-independent would be less confusing

Odd to say that NRPs are less modified than RiPPs when some have equal or greater complexity.

Figure 3: the color scheme here is non-sensical and some of the fonts are bordering on microscopic. Why is an alternative core finder needed only if LanC or YcaO are in the gene cluster? This caption is not written in an accessible fashion.

Page 13: the sequence given for lanthi-1794 is clearly incorrect, since lanthipeptides require Cys. This is probably the sequence for lasso-1795. This paragraph bounces around between lasso and lanthi, the narrative hard to follow (illustrative of many places in the text).

Figure 4: how can you call this "discovery" of radamycin when this is a known compound? Further, throughout the manuscript, the lasso peptides are referred to as novel, but how can this be true when they are previously reported?

Figure 7c/d: I do not see the transporter genes that are indicated in the legend.

What is the URL for the hypoRiPPAtlas server? I can see it listed in the GitHub but I don't believe it is listed in the manuscript.

Supp Fig 6: Caption says lasso peptide and lanthipeptide. Seems like an error.

Supp Fig 7: the mass accuracies here for the MS/MS are too variable to be correct. Some ions are most likely misassigned.

It would be delightful if all of the chemical structure drawings were as nice as what is shown on pages 25, 28, etc of the SI pdf. Please redraw all of the others using this style.

Supp Table 6: this table is an utter mess. Many of the RiPPs require multiple enzymes for formation, and the authors only list one, often not the class-defining enzyme, so it's confusing from the start. Next, lanthipeptide A, B, etc. are not recognized RiPP classes. There are subclasses of lanthipeptides but they are numbered, not letters. The number of remaining errors is staggering.

Goadsporin is not a cyanobactin

Cypemycin is not a lanthipeptide

Thioviridamide is not a linaridin

hominicin is not a lanthipeptide

There are probably more errors than this.

Reviewer #4 (Remarks to the Author):

The manuscript by Lee et. al., addresses a bottleneck in natural product research: connecting the huge amount of biosynthetic gene clusters detected by genome mining with the produced compounds. "HypoRiPPAtlas: an Atlas of hypothetical natural products for mass spectrometry database search" introduces a pipeline that identifies RiPP precursor genes within genomes and predicts the putatively encoded structures and fragmentation patterns. These can be subsequently identified in actual MS spectra and databases. This way the authors were able to identify several novel RiPPs from bacteria and plants showcasing the benefit of the pipeline. I believe that this manuscript is of high interest and high quality. After carefully reading the previous comments of the reviewers and point-by-point answers, I believe that the manuscript is ready to publish now.

Dear Dr Bratovic

Thank you for forwarding us very insightful reviews. Below please find answers to all questions raised by the reviewers.

I attached the revised manuscript as well as the detailed point-by-point answers to the reviews. We significantly revised and extended the paper to address the reviewers' questions (most additions were delegated to the Supplementary material to limit the length of the main manuscript). The changes/additions are shown in red.

Please see our responses below.

Hosein Mohimani

REVIEWER COMMENTS

Reviewer #1

The authors have carefully addressed all my comments from the previous round of review. I have no further comments or suggestions.

Reviewer #3

C3.1. This manuscript from the Mohimani lab describes HypoRiPPAtlas, which is an algorithm that links RiPP genome mining with the global natural products social molecular networking program. This is a novel and important concept, given that the latter houses a large amount of mass spectrometry data for natural products from various biological extracts. Based on a reading of the prior reviews, the authors have improved some aspects of the paper, but unfortunately, the paper is not ready for publication. Some rather egregious omissions and incorrect statements in the original manuscript appear to have been corrected, but some new text sections range from misleading to factually incorrect. Equally troubling is that the text itself is meandering, disorganized, and of unjustifiable length. The narrative is often confusing and some of the sections seem chronologically out of place. With respect to the figures, the presentation is below average. Based on several factors, most notably the large number of errors, the authors give the impression they do not know the RiPP field and haven't read the papers in their own citation list. A major rewrite is necessary before this manuscript should be published in any journal, let alone a multidisciplinary journal.

R3.1. Thank you for your suggestions. We have incorporated your recommendations into the manuscript. The revisions have been highlighted in red.

Major comments

C3.2. Page 3: To call GNPS a "gold mine" equates to hyperbole. In my experience, this database is not particularly well suited for RiPP discovery and the quality of the MS data is so highly variable that it often precludes usage.

R3.2. Thank you for the suggestion; we have removed the “gold mine” description and updated the paragraph as follows:

Since 2015, global natural product social (GNPS) molecular networking infrastructure has brought together over two thousand mass spectral datasets from over five hundred principal investigators containing over seven hundred thousand samples obtained from microbial isolates, host-oriented and environmental communities [14]. Accompanied with molecular networking [15] (a network of mass spectra, where similar spectra are connected with an edge), GNPS is a valuable resource for future natural product discovery. However, over 98% of the billion mass spectra currently stored at GNPS represent the ‘dark matter of metabolomics’ [16] since all attempts to interpret them have been failed. This ‘dark matter’ likely consists of spectra of unknown molecules produced by BGCs encoded in existing genomic repositories.

C3.3. Page 5, elsewhere: Why do the authors state in multiple locations that RODEO is only applicable to lasso peptides? Have the authors read the numerous papers (from their own citation list) that show RODEO being used for other RiPP classes? I have never heard anyone else say that RODEO only works for lasso peptides.

R3.3. Thank you for bringing this up.

In fact, the title of the original manuscript that reports RODEO mentions that this is specifically developed for lasso-peptides:

A new genome-mining tool redefines the lasso peptide biosynthetic landscape
Jonathan I. Tietz,^{1,†} Christopher J. Schwalen,^{1,†} Parth S. Patel,¹ Tucker Maxson,¹ Patricia M. Blair,¹ Hua-Chia Tai,¹ Uzma I. Zakai,¹ and Douglas A. Mitchell^{1,2,3,*}

However, since then, follow-up manuscripts have expanded RODEO, namely RODEO2, to other classes of natural products presented in the manuscript below.

Bioinformatic mapping of radical SAM-dependent RiPPs identifies new C α , C β , and C γ -linked thioether-containing peptides
Graham A. Hudson,^{†§} Brandon J. Burkhardt,^{†‡§} Adam J. DiCaprio,^{†‡} Christopher J. Schwalen,[†] Bryce Kille,[†] Taras V. Pogorelov,^{†¶§} // and Douglas A. Mitchell^{†‡*}

According to the reviewer's comment, we have updated the following statement:

RODEO [19] and its updated version, RODEO2[cite Graham et al. 2019], predict precursor and core peptides using motif search and machine learning for lassopeptides, class I lanthipeptides, sactipeptides, and thiopeptides.

C3.4. I don't follow the logic progression of including SynBNPs in this paragraph. This is a really big pivot. One idea per paragraph, please. The same is true throughout the paper. This is a criticism to help the authors produce a more readable text.

R3.4. Thank you for your suggestions. We removed the following sentences from the paragraph.

~~Synthetic-Bioinformatic Natural Products (syn-BNPs) [27] have emerged as an alternative to isolation-based natural product discovery approaches. Using syn-BNPs researchers no longer have to isolate bioactive molecules to study their activity. However, the syn-BNP approach has largely focused on non-ribosomal peptides as they are less widely modified than RiPPs. Furthermore, seq2RiPP can serve as an upstream step to the syn-BNP approach, as it provides high quality annotations and verifies theoretical RiPPs via mass spectral search.~~

Instead, we are now mentioning syn-BNP approach in the first paragraph of the **Introduction** as follows:

The natural products of cultured microbes have served as a major source of lead compounds for antibiotics [4], drug [5], food preservative [6], and analgesic agent [7, 8] discoveries. However, novel antibiotics are needed to combat antibiotics resistance, and a continued focus on the abundant molecules from cultured microbes is ineffective due to high rates of rediscovery. Traditional approaches rely on repeated fractionation and bioactivity testing, followed by isolation and structure elucidation of the molecules of interest, which is a time-consuming and expensive process. ~~The Synthetic-Bioinformatic Natural Products (syn-BNPs) [cite <https://www.nature.com/articles/nchembio.2207>], proposed as an alternative strategy, relies on predicting chemical structures with existing bioinformatic tools, and thus, its effectiveness is constrained by the limitations of these tools.~~

C3.5. “non-class-specific”? Class-independent would be less confusing

R3.5. Thank you for the suggestion. We have replaced non-class-specific with class-independent in the following paragraph:

~~RiPPER [20] builds upon RODEO outputs to predict class-independent RiPP precursors by adding ORF prediction via a custom build of gene prediction software Prodigal [21].~~

C3.6. Odd to say that NRPs are less modified than RiPPs when some have equal or greater complexity.

R3.6. Thank you for the suggestion. We removed the paragraph as shown in R3.4.

C3.7.1 Figure 3: the color scheme here is non-sensical and some of the fonts are bordering on microscopic. Why is an alternative core finder needed only if LanC or YcaO are in the gene cluster? This caption is not written in an accessible fashion.

R3.7 We thank you for your suggestion in Figure 3. We replied to each comment below.

Figure 3: the color scheme here is non-sensical and some of the fonts are bordering on microscopic.

We increased the font size, remade the figure, and explained the color scheme in the updated figure caption. Also, we made an alternative version of the figure in black and white for the reviewer to choose from.

Why is an alternative core finder needed only if LanC or YcaO are in the gene cluster? This caption is not written in an accessible fashion.

We apologize for the typo in including LanC. We have rewritten the description in the figure caption. The alternative core finding is enabled for cyanobacteria BGCs (which contain the YcaO gene motif) and plants.

a.

b.

c.

Figure 3: Bgc2orf and orf2core models. As illustrated in (a) and (b), from left to right, the red peptide is a RiPP ORF, and the yellow section is the RiPP core. The green blocks are two 1D CNNs, and the purple blocks are bidirectional LSTM with a dense layer and a CRF layer in bgc2orf and orf2core, respectively. (a) Bgc2orf model is a binary classifier that computes the probability of a given ORF peptide sequence being a RiPP ORF. Bgc2orf model consists of (1) a padding process and an embedding layer (shown in blue), (2) two 1D CNNs (shown in green), and (3) a single layer bidirectional LSTM, a flattening layer, and a dense layer (shown in purple). The output is a probability and the default cutoff is 0.5. (b) The orf2core model shares a similar

architecture with *bgc2orf*. However, the flattening and dense layers are replaced with a conditional random fields layer (shown in purple), which predicts the probability of each amino acid is one of the < start >, < before >, < core >, < after >, < end > tokens. The *orf2core* model takes a RiPP ORF as input and identifies *k* N-terminal and *k* C-terminal cleavage sites given the predicted tokens, where *k* is a user-defined hyperparameter. N- and C-terminal cleavage sites are defined as the transition from < before > to < core > and from < core > to < after >, respectively. Then, cores are predicted based on the combination of N- and C-terminal cleavage sites. (c) An alternative core finder is used to search the repeated leader-follower patterns, which are highlighted in gray, and to identify the core sequence in the patterns, highlighted in yellow. The alternative core finding is enabled for cyanobacteria BGCs (which contain the *YcaO* gene motif) and plants.

a.

b.

c.

Figure 3: *Bgc2orf* and *orf2core* models. As illustrated in (a) and (b), from left to right, the dark gray peptide demonstrates the process of filtering ORFs and identifying the core sequence, which is labeled in light gray. (a) *Bgc2orf* model is a binary classifier that computes the

probability of a given ORF peptide sequence being a RiPP ORF. Moving from left to right, each ORF is assigned a probability, and those with probabilities higher than 0.5 pass the filter.

Bgc2orf model consists of a padding process and an embedding layer two 1D CNNs, a single layer bidirectional LSTM, a flattening layer, and a dense layer. (b) The orf2core model shares a similar architecture with bgc2orf. However, the flattening and dense layers are replaced with a conditional random fields layer, which predicts the probability of each amino acid being one of the < start >, < before >, < core >, < after >, < end > tokens. The orf2core model takes a RiPP ORF as input and identifies k N-terminal and k C-terminal cleavage sites given the predicted tokens, where k is a user-defined hyperparameter. N- and C-terminal cleavage sites are defined as the transition from < before > to < core > and from < core > to < after >, respectively. Then, cores are predicted based on the combination of N- and C-terminal cleavage sites. (c) An alternative core finder is used to search repeated (at least twice) leader-follower patterns and identify the core sequence in the patterns, highlighted in gray, and identify the core sequence in the patterns, which are enclosed in boxes. The alternative core finding is enabled for cyanobacteria BGCs (which contain the YcaO gene motif) and plants.

C3.8. Page 13: the sequence given for lanthi-1794 is clearly incorrect, since lanthipeptides require Cys. This is probably the sequence for lasso-1795. This paragraph bounces around between lasso and lanthi, the narrative hard to follow (illustrative of many places in the text).

R3.8. Thank you for pointing this out. We apologize for putting the inaccurate sequence, and subfigure label in the wrong order. The correct core sequence is fixed. And the lanthi-1749 is in 7c; and lasso-1795 is in 7b. Also, we change the order of the description and supplementary figure 6 and 7, which describes both lasso peptides first, and then the lanthipeptide. We have fixed it by moving lasso-1795 before lanthi-1794 in the main text as follows :

Lasso-1648 is identified from *Streptomyces* NRRL B-2660, containing a N-terminal macrolactam ring between N₁ and D₈ (Figure 7a and Supplementary Figure S5). Based on Seq2ripp predictions, the PTM is applied by Asn-synthetase (PF00733.18). Lasso-1795 is identified from *Streptomyces* NRRL B-2660 and WC-3560, containing a N-terminal macrolactam ring between Q₁ and D₈ (Figure 7b and Supplementary Figure S6). Lanthi-1794 is identified from *Streptomyces* WC-3904. A dehydroalanine (Dha) located at S₆, and three dehydrobutyrines (Dhb) located in T₂, T₁₀, T₁₅ are potentially connected to one of the cysteines in the core peptide, forming lanthionine (Lan) or methyl-lanthionine (MeLan) rings (Figure 7c and Supplementary Figure S7). The PTM is applied by Lantibiotic dehydratase (PF05147.6 and PF04738.6).

C3.9.1 Figure 4: how can you call this “discovery” of radamycin when this is a known compound?

R3.9.1 Thank you for pointing this out. We apologize for this typo, and we have fixed it in the figure 4 caption:

Figure 4: Rediscovery of radamycin by seq2ripp.

C3.9.2 Further, throughout the manuscript, the lasso peptides are referred to as novel, but how can this be true when they are previously reported?

R3.9.2 We thank the reviewer for bringing this up. We modified the sentence as follows:

The DTGHCSGVCTVLVCTVAVC core identified by seq2ripp for lanthi-1794 does not appear in the survey conducted by Walker et al. [47]. However, the precursors and cores for lasso-1648 and lasso-1795 were previously reported by RODEO [19]. Seq2ripp validated that these lasso peptides are expressed naturally by the producing microorganisms through mass spectral search.

C3.10. Figure 7c/d: I do not see the transporter genes that are indicated in the legend.

R3.10. We have removed transport-related genes from Figure 7d legend. In Figure 7 do not see transporter-related genes in the hypothetical BGC.

C3.11. What is the URL for the hypoRiPPAtlas server? I can see it listed in the GitHub but I don't believe it is listed in the manuscript.

R3.11. Thank you for pointing this out. We now add the link to the Data and code availability section.

Data and code availability. All the MS datasets are publicly available from the GNPS infrastructure (microbes: Supplementary Table S8; plant: MassIVE MSV000088918). All plant transcriptomics datasets are publicly available from Zenodo [76] (<https://zenodo.org/record/6253747>). The link to the HypoRiPPAtlas server is <https://hyporippatlas.npanalysis.org>.

Instructions for using HypoRiPPAtlas are available from <https://github.com/mohimanilab/seq2ripp>.

C3.12. Supp Fig 6: Caption says lasso peptide and lanthipeptide. Seems like an error.

R3.12. Thank you for pointing this out. We apologize for the typo. We have fixed the typo. Also, corresponding to R3.8, we changed Supplementary Figure 6 to Supplementary Figure 7.

Supplementary Figure 7: Characterization of lanthipeptide Lanthi-1794 from *Streptomyces rimosus* subsp. *rimosus* WC-3904. (A) LCMS analysis of extract from *Streptomyces rimosus* subsp. *rimosus* WC-3904 (MassIVE: MSV000083738 raw P263 R8). (B) Detected MS signal of lanthipeptide Lanthi-1794 from *Streptomyces rimosus* subsp. *rimosus* WC-3904. (C) MS/MS analysis of Lanthi-1794.

C3.13. Supp Fig 7: the mass accuracies here for the MS/MS are too variable to be correct. Some ions are most likely misassigned.

Thank you for pointing this out. We removed the ions with $\Delta m > 5$ ppm and updated the figure as shown below. (Corresponding to R3.8. We changed Supplementary Figure 7 to Supplementary Figure 6.)

Fragment	m(obs)	m(calc)	Δm [ppm]
y2 (1+)	221.0947	221.0955	-3.6
y3 (1+)	292.1313	292.1326	-4.5
y4 (1+)	391.1996	391.2010	-3.6
y5* (1+)	474.2372	474.2381	-1.9
b5* (1+)	496.1595	496.1609	-2.8
b6** (1+)	565.1804	565.1823	-3.4
b7** (1+)	622.2009	622.2038	-4.6
y7* (1+)	676.3132	676.3157	-3.7
y8* (1+)	789.3970	789.3998	-3.5
b9** (1+)	824.2783	824.2814	-3.7
y9* (1+)	888.4647	888.4682	-3.9
b10*** (1+)	907.3149	907.3185	-3.9
b11*** (1+)	1006.3833	1006.3869	-3.6
b12*** (1+)	1119.4689	1119.4710	-1.8
b14*** (1+)	1321.5461	1321.5486	-1.9
b15*** (1+)	1404.5817	1404.5856	-2.8
b17*** (2+)	1574.6912	1574.6864	3.1

MSV000083738_raw_P263_R8#12495-12837 RT: 36.98-37.73 AV: 3 NL: 2.03E5
F: FTMS + p ESI d Full ms2 897.89@hcd25.00 [123.33-1850.00]

* = -H₂O (18.0106 Da)

C3.14. It would be delightful if all of the chemical structure drawings were as nice as what is shown on pages 25, 28, etc of the SI pdf. Please redraw all of the others using this style.

R3.14. Thank you for your suggestion. We have remade all chemical structures as shown below:

Bacterial Genomes

Figure 2: Seq2ripp pipeline.

Figure 4: Rediscovery of radamycin by seq2ripp.

Figure 7.

Figure 5: Annotation of radamycin mass spectra from GNPS actinomycete dataset MSV000078937 using the Dereplicator+ model.

Supplementary Figure 1: Identification of grisemycin by seq2ripp.

Supplementary Figure 2: Annotation of grisemycin mass spectra based on Dereplicator+ model.

(a) genome2bgc

(b) bgc2orf

Precursor sequence: MKEQNSFNLLQEVTESELDLILGAKGGSGVIHTISHECNMNSWQFVFTCCS

leader

core

(c) orf2core

K1 G2 G3 S4 G5 V6 I7 H8 T9 I10 S11 H12 E13 C14 N15 M16 N17 S18 W19 Q20 F21 V22 F23 T24 C25 C26 S27

(d) core2ripp

(e) Biosynthesis

Dehydration
Lanthionine formation

Supplementary Figure 3: Identification of lactacin 481 by seq2ripp.

Supplementary Figure 4: Annotation of lactacin 481 mass spectra based on Dereplicator+ model.

Fragment (z)	m(obs)	m(calc)	Δm [ppm]
y1 (1+)	132.1020	132.1021	1.1
y2 (1+)	245.1860	245.1861	0.4
y3 (1+)	373.2446	373.2441	1.3
y4 (1+)	501.3396	501.3414	3.7
y5 (1+)	558.3610	558.3615	0.9
y6 (1+)	615.3825	615.3830	0.9
b7 (1+)	725.4315	725.4305	1.4
y7 (1+)	728.4665	728.4688	3.1
b8* (1+)	822.4484	822.4468	2
y8 (1+)	827.5350	827.5350	0
b9* (1+)	921.5157	921.5153	0.4
b10* (1+)	1034.5996	1034.5993	0.3
b11* (1+)	1091.6227	1091.6208	1.7
b12* (1+)	1148.6404	1148.6422	1.6
b13* (1+)	1276.7358	1276.7372	1.1
b14* (1+)	1404.7950	1404.7958	0.6
b15* (2+)	1517.8829	1517.8798	2
b16* (2+)	1630.9657	1630.9639	1.1

NRRL-colony-TargetedSIM-DDA-824 #452-467 RT: 4.03-4.07 AV: 7 NL: 5.82E6
F: FTMS + p ESI d Full ms2 824.9915@hcd25.00 [113.3333-1700.0000]

Supplementary Figure 5: Characterization of lasso peptide Lasso-1648 from *Streptomyces rimosus* subsp. *rimosus* NRRL-2660.

* = -H₂O (18.0106 Da)

Supplementary Figure 6: Characterization of lasso peptide Lasso-1795 from *Streptomyces rimosus* subsp. *rimosus* NRRL-2660.

Fragment	m(obs)	m(calc)	Δm [ppm]
y2 (1+)	221.0947	221.0955	-3.6
y3 (1+)	292.1313	292.1326	-4.5
y4 (1+)	391.1996	391.2010	-3.6
y5* (1+)	474.2372	474.2381	-1.9
b5* (1+)	496.1595	496.1609	-2.8
b6** (1+)	565.1804	565.1823	-3.4
b7** (1+)	622.2009	622.2038	-4.6
y7* (1+)	676.3132	676.3157	-3.7
y8* (1+)	789.3970	789.3998	-3.5
b9** (1+)	824.2783	824.2814	-3.7
y9* (1+)	888.4647	888.4682	-3.9
b10*** (1+)	907.3149	907.3185	-3.9
b11*** (1+)	1006.3833	1006.3869	-3.6
b12*** (1+)	1119.4689	1119.4710	-1.8
b14*** (1+)	1321.5461	1321.5486	-1.9
b15*** (1+)	1404.5817	1404.5856	-2.8
b17*** (2+)	1574.6912	1574.6864	3.1

MSV000083738_raw_P263_R8#12495-12837 RT: 36.98-37.73 AV: 3 NL: 2.03E5
F: FTMS + p ESI d Full ms2 897.89@hcd25.00 [123.33-1850.00]

* = -H₂O (18.0106 Da)

Supplementary Figure 7: Characterization of lasso peptide Lanthi-1794 from *Streptomyces rimosus* subsp. *rimosus* WC-3904.

Supplementary Figure 16: The list of plant RiPP modifications used in this study.

LPIIY (Elaeagnin)

TFYY (Skr-618)

QPFGVFSW (Jdi-931)

VPIFY (Sali-635)

QPFGVFAW (Jdi-931)

QYGTH (Mtr-630)

QVPIFY (Sali-746)

QLKVW (Cca-653)

FLLY (Skr-552)

GEVTSY (Skr-652)

FLLYPY (Skr-838)

ILLY (Skr-546)

ILLYPSY (Skr-863)

Supplementary Figure 17: Novel plant RiPPs predicted by plant-seq2ripp (Supplementary Table S3).

Supplementary Figure 19: Annotation of elaeagnin mass spectra from *Eleaagnus pungens* root using Dereplicate+ model.

C**D**
E**F**

Supplementary Figure 20: Structure elucidation of elaeagnin.

Supplementary Figure 21: A novel post-translational modification with a crosslink between the hydroxyl group of the tyrosine and the β -carbon of the proline.

A >EpuBURP (*Elaeagnus pungens*, ERR2040413, SPAdes 3.13)
 MELRLFLFLVLIHLGAVSSQFDDSHNLHNEGEEAFPNVDKLPQAGGVPEKFGKDLLNHMAQSGLFSGGVSEKFGKLLNHMAQPDIL
 PIIYGGGAHENFGKDLLNHMAQPDLLPIIYGGGAPEKIEKDLLNHKAQSDLFSGGASEKFGKDLLNHMAQPDMLPIIYGGGAHENFGKDLL
 NHMAQPDLLPIIYGGGAPEKIEKDLLNHKAQSDLFSGGASEKFGKLLNHMAQPNMLPIIYGGGAHENFGKLLNHMAQPDMLPIIYGGGA
 PMKDSKLMKGNVASFLENLLGKTMKLFHFNPIITGAKFLPLDIAKSI PFASNKLPEILNRFNIEPKSSDVEIVKQTI SLCEGQKSIETE
 HKYCATSLESLEIDFSRSLGEDIKIYSTEVDDEIKQDYKIIKESIVKLGDKSVVCHKLNMYAVVYCHHVHATKVYMATLEGENGVKENAI
 AACHADTKGWNPKHLAFQLLNIKPGTDTICHFLSSDTLVLVLPQKDKDVS

>EpuBURP
 ATGGAATTAGCTCCTTTTCCTTTTCTTCTCGTCATCTTGCATCTTGCAGGGGTGAGCAGCCAGTTTGTATGACAGTCACAACCTTCAACAAGGAGAGG
 AAGCCTTCCAAATGTTGACAAGCTCCCAACACAACAGCTGGTGGTGCCTGAAAAATTTGGAAAAGATCTTTTGAATCACAATGGCTCAATCTGGCTTATT
 TAGTGGTGGTGTCTGAAAAATTTGGAAAAATCTTTGAATCATATGGCTCAACCTGACATCTCCCTATAATTTATGGTGGAGTGCCCATGAAAAATTT
 GGAAAAGATCTTTTGAATCACAATGGCTCAACCTGACTTATTGCCTATAATTTATGGTGGTGGTGCCTGAAAAATTTGAAAAAGATCTTTTGAATCACAAGG
 CTCATCTGACTTATTAGTGGTGGTGCCTGAAAAATTTGGAAAAGATCTTTTGAATCATATGGCTCAACCTGACATGCTCCCTATAATTTATGGTGGAGG
 TGCCCATGAAAAATTTGGAAAAGATCTTTTGAATCACAATGGCTCAACCTGACTTATTGCCTATAATTTATGGTGGTGGTGCCTGAAAAATTTGAAAAAGAT
 CTTTGAACCACAAGGCTCAATCTGACTTATTAGTGGTGGTGCCTGAAAAATTTGGAAAAATCTTTTGAATCATATGGCTCAACCTAACATGCTCCCTA
 TAATTTATGGTGGAGTGCCCATGAAAAATTTGGAAAAATCTTTTGAATCACAATGGCTCAACCTGACATGCTCCCTATAATTTATGGTGGAGTGCCCAAT
 GAAAGATTCAAAATTAATGAAGGTAATGTTGCCCTTCTTCTTAGAGAACGATCTACTACTAGGTAAACAATGAAGTGCATTTTACAATCCCATAACT
 GGTGCTAAGTTCACCTCTTGATATGCAAAAATCCATACCTTTTGCAGTAACAAGTTACCAGAAAATTTTAAACCGTTTCAATATTGAGCCTAAATCATCTG
 ATGTAGAAAATTTAAAACAAAATATTAGTTTGTGTGAGGACAAAAAGCATTGAAACTGAACATAAGTATTGTGCAACATCACTCGAGTCATGATCGATTT
 TAGTAGGTCAAAACTTGGAGAAGACATAAAGATCTACTCAACAGAGGTGGATGAAGAAATCAAACAAGATTACAAAATCATTAAAGGAATCAATGTTAAGTTA
 GGAGACAAATCTGTGGTTTGCATAGCTAAATATATGTACGCTGTATATTTGTTCATCATGTTTCACTACAAAGGTGTATATGGCTACATTAGAAGGTG
 AAAATGGGGTAAAAGAAAATGCAATAGCTGCTTGTATGCTGATACTAAAGGTTGGAATCCTAAGCATTAGCTTTCAACTCCTCAATATCAAGCTGGAAC
 AGATACTATTGCCACTTCTTCAAGTGACACCCTTGTGGTTCCTCAAAGGACAAGGATGTTAGTTATTGA

B >EpuBURP-truncated (5xLPIIY)
 MELRLFLFLVLIHLGAVSSQFDDSHNLHNEGEEAFPNVDKLPQAGGVPEKFGKDLLNHMAQSGLFSGGVSEKFGKLLNHMAQPDIL
 PIIYGGGAHENFGKDLLNHMAQPDLLPIIYGGGAPEKIEKDLLNHKAQSDLFSGGASEKFGKDLLNHMAQPDMLPIIYGGGAHENFGKDLL
 NHMAQPDLLPIIYGGGAPMKDSKLMKGNVASFLENLLGKTMKLFHFNPIITGAKFLPLDIAKSI PFASNKLPEILNRFNIEPKSSDVEI
 VKQTI SLCEGQKSIETECHKYCATSLESLEIDFSRSLGEDIKIYSTEVDDEIKQDYKIIKESIVKLGDKSVVCHKLNMYAVVYCHHVHATK
 VYMATLEGENGVKENAIAACHADTKGWNPKHLAFQLLNIKPGTDTICHFLSSDTLVLVLPQKDKDVS

Supplementary Figure 22: Ribosomal biosynthesis of elaeagnin.

Supplementary Figure 24.

C3.15. Supp Table 6: this table is an utter mess. Many of the RiPPs require multiple enzymes for formation, and the authors only list one, often not the class-defining enzyme, so it's confusing from the start. Next, lanthipeptide A, B, etc. are not recognized RiPP classes. There are subclasses of lanthipeptides but they are numbered, not letters. The number of remaining errors is staggering.

Goadsporin is not a cyanobactin

Cypemycin is not a lanthipeptide

Thioviridamide is not a linardin

hominicin is not a lanthipeptide

There are probably more errors than this.

R3.15. Thank you for pointing this out. We apologize for the errors we made. We have carefully gone through the table and fixed the errors. The RiPP class has been corrected based on the class listed on MIBiG, RiPPMiner, on April 8, 2023, or the original paper.

Regarding the comment, “Many of the RiPPs require multiple enzymes for formation, and the authors only list one, often not the class-defining enzyme.”:

Supplementary Table 6 is not intended to be a comprehensive summary of all modifications occurring in each RiPP. Rather, it represents a list of post-translational modifications (PTMs) implemented in seq2ripp. We collected the list from Arnison *et al.* (2013) [7], Montalbán-López *et al.* (2021) [68], and our in-house database. The second column shows one example of RiPP possessing that particular PTM. The third column shows the peptide sequence of that RiPP. To clarify this point, we have modified the second column's title from "peptide name" to "an example peptide name." Furthermore, we have revised the table caption and description in the main text as follows:

Supplementary Figure S24 and Supplementary Table S6 summarize all the modifications implemented in seq2ripp that we collected from Arnison *et al.* (2013) [7], Montalbán-López *et al.* (2021) [68], and our in-house database [18]. In Supplementary Table S6, we present one RiPP example for each modification.

mod	example peptide name	peptide sequence	name of enzyme	class
F+68	Aeruginosamide C	FFPVC	PF08241 Methyltrans_11	Cyanobactin
Y+68	Aesturamides7-12	ACMPCYP	PF08241 Methyltrans_11	Cyanobactin
S+68	Trunkamide	TSIAPFC	PatF	Cyanobactin
T+68	Trunkamide	TSIAPFC	PatF	Cyanobactin
C-6.cter	Aeruginosamide C	FFPVC	TsrB	Cyanobactin
C-18	Aerucyclamide A	ITGCIC	LazE (PF02624)	Cyanobactin
T-20	Radamycin	SCVGSACACSSSSSS	McbC/D (PF02624)	Thiopeptide
S-20	plantazolicin A	RCTCTTHSSSTF	McbC/D (PF02624)	LAP
M+28	venepptide	MNVITNLLAGVVHPLGWLV	Non-enzymatic	N-formylated
C+162	Sublancin 168	GLGKAQCAALWLQCASGGTIGCGGGAVA CQNYRQFCR	Glycos-transf-2 (PF00535)	Lanthipeptide
S-18	Radamycin	SCVGTACACSSSTSSS	LanB/M (PF05147)	Thiopeptide
S-17	pinensin A	SHPHTVATDDQGHLCCTHCA	SSF51735	Thiopeptide
T-17	Pep5	TAGPAIRASVKQCQKTLKATRLFTVSCKG KNGCK	SSF51735	Lanthipeptide
S-15	Epilancin 15X	SASIVKTTIKASKKLCRGFTLTCGCHFTG KK	ElxO (SSF51735)	Lanthipeptide
CXC-80	cypemycin	ATPATPTVAQFVIQGSTICLVC	Flavoprotein (PF02441)	Linaridin
SXC-64	epidermin	IASKFICTPGCAKTGSFNSYCC	Flavoprotein (PF02441)	Lanthipeptide
TXC-64	epidermin	IASKFICTPGCAKTGSFNSYCC	Flavoprotein (PF02441)	Lanthipeptide
S-16	Lacticin 3147 A1	CSTNTFSLSDYWGNGAWCTLTHECMAW CK	LtnJ	Lanthipeptide
D+16	cinnamycin	CRQSCSFGPFTFVCDGNTK	McbC/D (PF02624)	Lanthipeptide
C-1	methanobactin	LCGSCYPCSCM	DsbB (PF02600)	Methanobactin
W+34	microbisporicin A1	VTWSLCTPGCTSPGGGNSCSFCC	MibH (PF04820)	Lanthipeptide
P+16	phalloidin	AWLATCP	MibO (PF00067)	Phallotoxins
P+32	microbisporicin A1	VTWSLCTPGCTSPGGGNSCSFCC	MibO (PF00067)	Lanthipeptide
SXC+16	Actagardine	SSGWVCTLTIECGTVICAC	GarO (PF00296)	Lanthipeptide
TXC+16	Actagardine	SSGWVCTLTIECGTVICAC	GarO (PF00296)	Lanthipeptide
A+42	microviridin J	ISTRKYPSDVEEW	mdnD	Microviridin
T-18	Lacticin 3147 A1	CSTNTFSLSDYWGNGAWCTLTHECMAW CK	LanB/M (PF05147)	Lanthipeptide
T-16	Nisin A	ITSISLCTPGCKTGALMGCNMKTATCHCS IHVSK	FMN-red (PF03358)	Lanthipeptide
C-20	Radamycin	SCVGTACACSSSTSSS	McbC/D (PF02624)	Thiopeptide
R+28	plantazolicin A	RCTCTTHSSSTF	Methyltransf-31 (PF13847)	LAP
CXD-18	Siamycin I	CLGVGSCNDFAGCGYAIUCFW	MecJ	Lasso peptide
CXE-18	Siamycin I	CLGVGSCNDFAGCGYAIUCFW	MecJ	Lasso peptide
GXD-18	RES-701-1	GNWHGTAPDWFFFNYYW	MecJ	Lasso peptide
GXE-18	Lariatrin A	GSQLVYREWVGHNSVIKPF	MecJ	Lasso peptide
A+28	Cypemycin	IASKFICTPGCAKTGSFNSYCC	CypM (PF13649)	Linaridin
I+28	Cypemycin	IASKFICTPGCAKTGSFNSYCC	CypM (PF13649)	Linaridin
L+28	Cypemycin	IASKFICTPGCAKTGSFNSYCC	CypM (PF13649)	Linaridin
C-6.pyrazinedione	methanobactin	LCGSCYPCSCM	mbnB/C/E/H/S	Methanobactin
C-4	methanobactin	LCGSCYPCSCM	mbnB/C/E/H/S	Methanobactin
L-1	methanobactin	LCGSCYPCSCM	unknown enzyme	Methanobactin
T+80	methanobactin	RCASTCAATNG	mbnS	Methanobactin
T+39	Polytheonamides	TGIGVVAVVAGAVANTGAGVNQVAGG NINVVGNINVNANVSVNMQTT	PoyF (PF13575)	Proteusins
M+44	Polytheonamides	TGIGVVAVVAGAVANTGAGVNQVAGG NINVVGNINVNANVSVNMQTT	B12-binding (PF01497)	Proteusins
I+14	Polytheonamides	TGIGVVAVVAGAVANTGAGVNQVAGG NINVVGNINVNANVSVNMQTT	B12-binding (PF01497)	Proteusins
Q+14	Polytheonamides	TGIGVVAVVAGAVANTGAGVNQVAGG NINVVGNINVNANVSVNMQTT	B12-binding (PF01497)	Proteusins
V+14	Polytheonamides	TGIGVVAVVAGAVANTGAGVNQVAGG NINVVGNINVNANVSVNMQTT	Radical.SAM (PF04055)	Proteusins
T+14	Polytheonamides	TGIGVVAVVAGAVANTGAGVNQVAGG NINVVGNINVNANVSVNMQTT	B12-binding (PF01497)	Proteusins
N+14	Polytheonamides	TGIGVVAVVAGAVANTGAGVNQVAGG NINVVGNINVNANVSVNMQTT	MTS (PF05175)	Proteusins
N+16	Polytheonamides	TGIGVVAVVAGAVANTGAGVNQVAGG NINVVGNINVNANVSVNMQTT	Cupin-4 (PF08007)	Proteusins
N+30	Polytheonamides	TGIGVVAVVAGAVANTGAGVNQVAGG NINVVGNINVNANVSVNMQTT	Cupin-4 (PF08007)	Proteusins
V+16	Polytheonamides	TGIGVVAVVAGAVANTGAGVNQVAGG NINVVGNINVNANVSVNMQTT	TpdJ1/2 (PF00067)	Proteusins
SXC-Lan/TXC-MeLan	Nisin A	ITSISLCTPGCKTGALMGCNMKTATCH CSIHVSK	LANC.like (PF05147)	Lanthipeptide
SXS-pyridine	Radamycin	SCVGTACACSSSTSSS	LazC (PF14028)	Thiopeptide
N-homolog	Radamycin	SCVGTACACSSSTSSS	kocH	Thiopeptide
Cter-1	Siomycin A	VSSASCTTCICTCSCSS	TsrC (PF00733)	Thiopeptide
S-20-zoline	goadsporin	ATVSTILCSGGTLSSAGCV	YcaO (PF02624)	LAP
T-20-zoline	goadsporin	ATVSTILCSGGTLSSAGCV	YcaO (PF02624)	LAP
C-20-zoline	patellamide A	ITVCISVC	YcaO (PF02624)	Cyanobactin
F-4	YM 216391	FIVGSSSC	Maf (PF02545)	YM-216391 family peptides
C-6.thia	thiomuracin A	SCNCFYICCCSSSA	Radical.SAM (PF04055)	Thiopeptide
C+24	GE2270	SCNCVCGFCCSCSPSA	Methyltransf 2 (PF00891)	Thiopeptide

mod	example peptide name	peptide sequence	name of enzyme	class
(S/T)X(S/T)XC	catenulipeptin	GHGGGGDSGLSVTGCNGHSGISLLCDL	Pkinase (PF00069)	Lanthipeptide
SXS-pyridine_hydroxy	nocathiacin	SCTTCECSCSCSS	Cytochrome_CBB3 (PF13442)	Thiopeptide
SXS_dehydropiperidine	siomycin A	VSSASCTTCICTCSCSS	Adh_short (PF00106)	Thiopeptide
SXS_piperidine	thiopeptin	VASASCTTCICTCSCSS	NadA (PF02445)	Thiopeptide
macrolacamide	bottromycin A2	GPVVVFDC	Peptidase_M27 (PF01742)	Thiopeptide
CX(D/E)+W	nosiheptide	SCTTCECSCSCSS	Abhydrolase6 (PF12697)	Thiopeptide
TXX+W	Thiostrepton	IASASCTTCICTCSCSS	Aminotran_1.2 (PF00155)	Thiopeptide
(S/T)_glycosylation	glycocin F	KPAWCWYTLAMCGAGYDSGTCDYMYSH	Endonuclease_NS (PF01223)	Sactipeptides
		CFGIKHHSSGSSSYHC		
C_glycosylation	Sublancin 168	GLGKAQCAALWLQCASGGTIGCGGGA	Glycos_transf_2 (PF00535)	Lanthipeptide
		CQNYRQFCR		
CXX-2	subtilosin A	NKGCATCSIGAAACLVDGPIPDFEAGATGL	Alba (PF01918)	Thiopeptide
		FGLWG		
(D/E)_Lasso	capistruin	GTPGFQTPDARVISRFGFN	Asn_synthase (PF00733)	Lassoepitide
AXC-2	cyclothiazomycin	SNCTSTGTPASCCSCCCC	CltM	Thiopeptide
N+343	microcin C7	MRTGNAD	ThiF (PF00899)	Microcin
X+16	thioviridamide	SVMAAAASIALHC	YcaO (PF02624)	Linaridin
W+123	comX	ADPITRQWGD	polyprenyl_synt (PF00348)	ComX
Cter+14	Aeruginosamide C	FFPVC	T4SS_CagC (PF16943)	Cyanobactin
CXcter	Agr autoinducing peptide	YSTCDFIM	AgrB (PF04647)	Autoinducing peptide
Cter+70	hominicin	ITPATPFTPAITEITA AVIAX	Unknown	Lanthipeptide
T-46	TP-1161	SCTTTGCACSSSSST	Fe-ADH (PF00465)	Thiopeptide
Nter+68	Aeruginosamide C	FFPVC	DHBP_synthase (PF00926)	Cyanobactin
Nter+42	goadsporin	ATVSTILCSGGTLSSAGCV	Acetyltransf_3 (PF13302)	LAP
H+30	thioviridamide	SVMAAAASIALHC	SAM-dependent_Mtases (IPR029063)	Linaridin
T+12	nocathiacin	SCTTCECSCSCSS	BPD_transp_1 (PF00528)	Thiopeptide
D-14	bottromycin A2	GPVVVFDC	LCM (PF04072)	Thiopeptide
T-14_ether	thiocillin	SCTTCVCTCSCCTT	Methyltransf_11 (PF08241)	Thiopeptide
I+14_ethylene_oxide	thiomuracin A	SCNCFYICCCSSA	p450 (PF00067)	Thiopeptide
I+14_pyrrrolidinol	GE 37468	STNCCXCYICCCSSN	p450 (PF00067)	Thiopeptide
KXW-2	streptide	AKGDGKVM	Carb_kinase (PF01256)	Unknown
KX(D/E)-3	microviridin B	FGTTLKYPDWEY	RimK (PF08443)	Microviridin
head-to-tail cyclization	patellamide A	ITVCISVC	PGM_PMM_I (PF02878)	Cyanobactin
I+32	siomycin A	VSSASCTTCICTCSCSS	p450 (PF00067)	Thiopeptide
6n-nitrogen-heterocycle	siomycin A	VSSASCTTCICTCSCSS	adh_short (PF00106)	Thiopeptide
P+16v2	Bottromycin A2	GPVVVFDC	btmC	Bottromycin
F+14	Bottromycin A2	GPVVVFDC	btmC	Bottromycin
S-13	Epicidin 280	SLGPAIKATRVCPKATRFVTVSCKKSDC	eciO	Lanthipeptide
		Q		
T-0bu	pep5	TAGPAIRASVKQCQKTLKATRLFTVSCKG	L_biotic_typeA (PF04604)	Lanthipeptide
		KNGCK		
S-pry	plantaricin W beta	SGIPCTIGAAVAASIAVCPPTTKSKRCGR	L_biotic_typeA (PF04604)	Lanthipeptide
		KK		
CXF-2	subtilosin A	NKGCATCSIGAAACLVDGPIPDFEAGATGL	PqqD (PF05402)	Thiopeptide
		FGLWG		
CXT-2	subtilosin A	NKGCATCSIGAAACLVDGPIPDFEAGATGL	PqqD (PF05402)	Thiopeptide
		FGLWG		
CXS-2	subtilosin A	NKGCATCSIGAAACLVDGPIPDFEAGATGL	PqqD (PF05402)	Thiopeptide
		FGLWG		
CXA-2	subtilosin A	NKGCATCSIGAAACLVDGPIPDFEAGATGL	PqqD (PF05402)	Thiopeptide
		FGLWG		
CXM-2	subtilosin A	NKGCATCSIGAAACLVDGPIPDFEAGATGL	PqqD (PF05402)	Thiopeptide
		FGLWG		
DXS-18	microviridin B	FGTTLKYPDWEY	MvdD (IPR026439)	Microviridin
DXT-18	microviridin B	FGTTLKYPDWEY	MvdD (IPR026439)	Microviridin
ExS-18	microviridin B	FGTTLKYPDWEY	MvdD (IPR026439)	Microviridin
EXT-18	microviridin B	FGTTLKYPDWEY	MvdD (IPR026439)	Microviridin
T-45-cter	micrococin P1	SCTTCVCTCSCCTT	adh_short (PF00106)	Thiopeptide
Cter_dehydr	enterocin A	MAKEFGIPA AVAGTVLNVVEAGGWVTTI	Unknown	Head-to-Tail Cyclized
		VSILTAVGSGGLSLLAAAGRE-SIKAYLKKEIKKGRVIAW		
R+1	citralassin	LLGLAGNDRVLVLSKN	Unknown	Lassoepitide
KXS-18	cinnamycin	CRQSCSFGPPTFVCDGNTK	durN	Lanthipeptide
KXT-18	cinnamycin	CRQSCSFGPPTFVCDGNTK	durN	Lanthipeptide
W+68	kawaguchipectin A	WLNQDNNWSTP	kgpF	Cyanobactin
SQ-amide	klebsazolicin	SQSPGNCASCSNSASANCTGGLG	mcbB	LAP
Y-proteusin	pcpA	VTAVGGVTGSGGIYGFPIQAMYGAVVGD	pcpX	Proteusin
		KPGKDWGWRFPSPLPKPSPIPS- WKPPV DVQPMYGVVSNDS		
S+41	thioviridamide	SVMAAAASIALHC	L_biotic_typeA (PF04604)	Linaridin
H+44	thioviridamide	SVMAAAASIALHC	tvag	Linaridin
H+14	microcyclamide	AFDGDEAS	Unknown	Cyanobactin
F+16	thiomuracin A	SCNCFYICCCSSA	MibO (PF00067)	Thiopeptide

Supplementary Table 6: The list of RiPP modifications considered in this study, along with a peptide example, the genes responsible for each modification, and their class. The chemical modifications are visualized in Supplementary Figure 22. (X = any amino acid)

Reviewer #4

The manuscript by Lee et. al., addresses a bottleneck in natural product research: connecting the huge amount of biosynthetic gene clusters detected by genome mining with the produced

compounds. "HypoRiPPAtlas: an Atlas of hypothetical natural products for mass spectrometry database search" introduces a pipeline that identifies RiPP precursor genes within genomes and predicts the putatively encoded structures and fragmentation patterns. These can be subsequently identified in actual MS spectra and databases. This way the authors were able to identify several novel RiPPs from bacteria and plants showcasing the benefit of the pipeline. I believe that this manuscript is of high interest and high quality. After carefully reading the previous comments of the reviewers and point-by-point answers, I believe that the manuscript is ready to publish now.

REVIEWERS' COMMENTS

Reviewer #3 (Remarks to the Author):

SI table 6 near end of doc remain:

Subtilosin is not a thiopeptide, it is a sactipeptide.

Sublancin is not a lanthipeptide

Thioviridamide is not a linaridin

I imagine more errors remain in this table.

Rodeo classifies more RiPPs than stated, even in this somewhat corrected document. A quick visit to the webtool lists all of the classes, as well as a quick search of the literature

REVIEWERS' COMMENTS

Reviewer #3 (Remarks to the Author):

C3.1 SI table 6 near end of doc remain:

Subtilosin is not a thiopeptide, it is a sactipeptide.

Sublancin is not a lanthipeptide

Thioviridamide is not a linaridin

I imagine more errors remain in this table.

R3.1 Thank you for pointing out this error. We have updated the SI Table 6 according to the information in Montalbán-López, M. et al. 2021.

C3.2 Rodeo classifies more RiPPs than stated, even in this somewhat corrected document. A quick visit to the webtool lists all of the classes, as well as a quick search of the literature

R3.2 We have updated the following statement based on the reviewer's comment:

RODEO [20] and its updated version, RODEO2[21], predict precursor and core peptides using motif search and machine learning for lassopeptides, class I-IV lanthipeptides, sactipeptides/ranthipeptide, graspetide, linaridin, pyritide, and thiopeptides.

Reference:

[1] Montalbán-López, M. et al. New developments in RiPP discovery, enzymology and engineering. Natural Product Reports vol. 38 130–239 (2021).